# Self-other generalisation shapes social interaction and is disrupted in borderline personality disorder

Joseph M Barnby[1,2,3]*, Jen Nguyen[1], Julia Griem[4,5], Magdalena Wloszek[5], Henry Burgess[6], Linda J Richards[6], Jessica Kingston[1], Gavin Cooper[1], P Read Montague[7], Peter Dayan[8,9], Tobias Nolte[4,5], Peter Fonagy[4,5], London Personality and Mood Disorders Consortium

[1]Department of Psychology, Royal Holloway, University of London, London, United Kingdom; [2]Institute of Psychiatry, Psychology and Neuroscience, King's College London, London, United Kingdom; [3]Centre for AI and Machine Learning, Edith Cowan University, Perth, Australia; [4]Department for Clinical, Educational, and Health Psychology, Division of Psychology and Language Sciences, University College London, London, United Kingdom; [5]Anna Freud, London, United Kingdom; [6]Department of Neuroscience, Washington University in St. Louis, St Louis, United States; [7]Centre for Human Neuroscience Research, Virginia Tech, Blacksburg, United States; [8]Max Planck Institute of Biological Cybernetics, Tübingen, Germany; [9]University of Tübingen, Tübingen, Germany

*For correspondence:
joseph.barnby@kcl.ac.uk

Competing interest: The authors declare that no competing interests exist.

## eLife Assessment

The findings are **important** and intriguing, with theoretical or practical implications beyond a single subfield. The computational methods employed are clever and sophisticated and the strength of evidence is **convincing**. Both the hypotheses and the exploratory nature of additional analyses are clearly stated.

**Abstract** Generalising information from ourselves to others, and others to ourselves allows for both a dependable source of navigation and adaptability in interpersonal exchange. Disturbances to social development in sensitive periods can cause enduring and distressing damage to lasting healthy relationships. However, identifying the mechanisms of healthy exchange has been difficult. We introduce a theory of self-other generalisation tested with data from a three-phase social value orientation task – the Intentions Game. We involved humans with (*n*=50) and without (*n*=53) a diagnosis of borderline personality disorder and assessed whether infractions to self-other generalisation may explain prior findings of disrupted social learning and instability. Healthy controls initially used their preferences to predict others and were influenced by their partners, leading to self-other convergence. In contrast, individuals with borderline personality disorder maintained distinct self-other representations when learning about others. This allowed for equal predictive performance compared to controls despite reduced updating sensitivity. Furthermore, we explored theory-driven individual differences underpinning contagion. Overall, the findings provide a clear explanation of how self-other generalisation constrains and assists learning, and how childhood adversity is associated with separation of internalised beliefs. Our model makes clear predictions about the mechanisms of social information generalisation concerning both joint and individual reward.

## Introduction

Social animals have evolved sophisticated mechanisms for cooperation, exchanging information to enable both individual and group regulation (*Emerson, 1956*; *Wheeler, 1911*). In humans, such exchanges provide crucial insights about others, as well as oneself, fostering the development of representations encoding self and others that is fundamental for adaptive social orientation and interaction. Disruptions to this process can impair the formation of stable social bonds and result in rigid interpersonal beliefs (*Fairbairn, 1952*; *Young et al., 2006*).

To effectively navigate what is sometimes called relational uncertainty, individuals generalise information from themselves to others (self-to-other transfer) and from others to themselves (other-to-self transfer). The relational self theory (*Andersen and Chen, 2002*) posits that when uncertain about others' behaviours, people often rely on their own preferences as initial hypotheses, a process termed self-insertion (*Allport, 1924*; *Krueger and Clement, 1994*). Conversely, when uncertain about their own states, individuals use external social cues to adjust their self-representations, a process known as social contagion (*Deutsch and Gerard, 1955*; *Toelch and Dolan, 2015*; *Moutoussis et al., 2016*). This bidirectional generalisation has been widely observed across various domains, including economic decision-making, morality, and social group adaptation (*Devaine and Daunizeau, 2017*; *Garvert et al., 2015*; *Panizza et al., 2021*; *Suzuki et al., 2016*; *Yu et al., 2021*).

Understanding healthy interpersonal dynamics can be clarified by examining their disruptions. Individuals with borderline personality disorder (BPD) provide a compelling case study due to their profound interpersonal instability, emotional dysregulation, and heightened sensitivity to social contexts (*Gunderson et al., 2018*). BPD is strongly associated with adverse childhood experiences (*Afifi et al., 2011*), impaired mentalising abilities (*Fonagy and Luyten, 2009*), and maladaptive representations of self and other (*Hanegraaf et al., 2021*). This phenotype has been explained through disrupted and unstable social inference during observation (*Henco et al., 2020*; *Hula et al., 2018*; *Story et al., 2024b*; *Siegel et al., 2020*). However, the precise mechanisms linking disrupted social cognition in BPD remain elusive, particularly regarding whether individuals with BPD differ specifically in their use of self-to-other and other-to-self information transfer.

This paper seeks to address this gap by testing explicitly how disruptions in self-other generalisation processes may underpin interpersonal disruptions observed in BPD. Specifically, our hypotheses were: (i) healthy controls will demonstrate evidence for both self-insertion and social contagion, integrating self and other information during interpersonal learning; and (ii) individuals with BPD will exhibit diminished self-other integration, reflected in stronger evidence for observations that assume distinct self-other representations.

We tested these hypotheses by designing a dynamic, sequential, three-phase social value orientation (*Murphy and Ackermann, 2014*) paradigm – the Intentions Game – that would provide behavioural signatures assessing whether BPD differed from healthy controls in these generalisation processes (*Figure 1A*). We coupled this paradigm with a lattice of models (M1-M4) that distinguish between self-insertion and social contagion (*Figure 1B*) and performed model comparison:

> M1. Both self-to-other (self-insertion) and other-to-self (social contagion) occur before and after learning
> M2. Self-to-other transfer only occurs
> M3. Other-to-self transfer only occurs
> M4. Neither transfer process, suggesting distinct self-other representations

We additionally ran exploratory analysis of parameter differences and model predictions between groups following from prior work, demonstrating changes in prosociality (*Hula et al., 2018*), social concern (*Henco et al., 2020*), belief stability (*Story et al., 2024b*), and belief updating (*Story et al., 2024a*) in BPD to understand whether discrepancies in self-other generalisation influence observational learning. By clearly articulating our hypotheses, we aim to clarify the theoretical contribution of our findings to existing literature on social learning, BPD, and computational psychiatry.

## Results

Healthy participants (CON; n=53) and participants diagnosed with BPD (n=50), matched on age, gender, education, and social deprivation indices (*Table 1*), were invited to participate in a three-phase

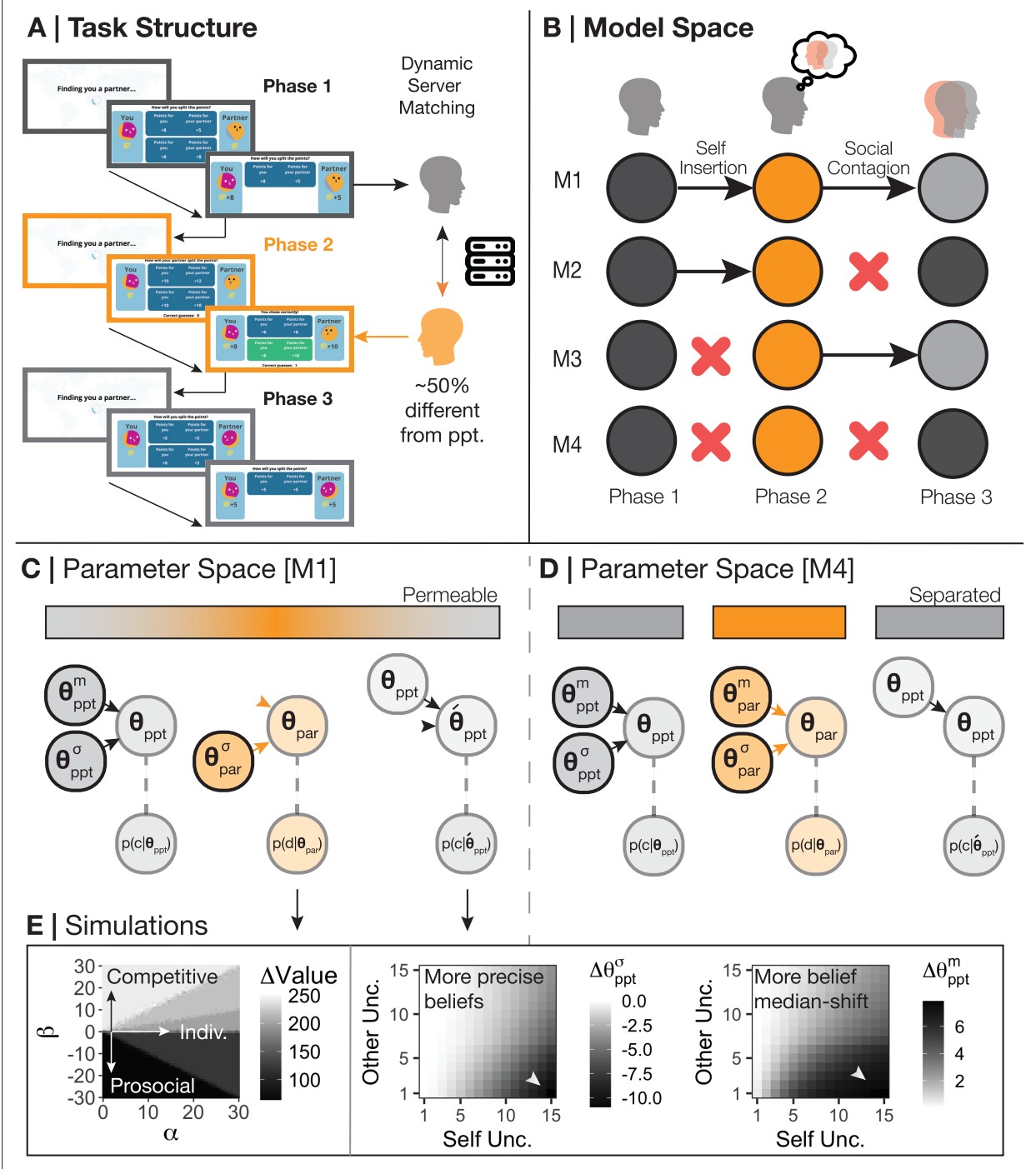

**Figure 1.** Task and model space. (**A**) Participants were invited to play a three-phase, repeated social value orientation paradigm – the Intentions Game – with virtual partners. Phase 1 of the Intentions Game lasted 36 trials and asked participants to make a forced choice between two options as to how to split points with an anonymous virtual partner. An example of a prosocial-individualistic pair of options could be (self = 5, other = 5) or (self = 10, other = 5) – if the participant chooses option 1, they could be viewed as less individualistic and more prosocial as the outcomes to the other do not change, but the self would earn less. In phase 2, lasting 54 trials, participants were asked to predict the decisions of a new anonymous partner using the same two-forced choice set-up and the same option pairs; participants were given feedback on whether they were correct or incorrect in their prediction. We used Amazon Web Services to create a novel server architecture to match participants and (virtual) partners (**Burgess and Barnby, 2023**). Partners in phase 2 were matched to be approximately 50% different from the participant with respect to their choices in phase 1 to ensure all participants needed to learn about their phase 2 partner, and to provide a mechanism to examine whether beliefs about partners had an effect on the self. Phase 3 was identical to phase 1, although participants were informed that they were matched with a third anonymous partner, unconnected to the partners in phases 1 and 2. At the end of the game, if participants collected over 1000 points overall, they were entered into a lottery to win a bonus. (**B**) We created four

*Figure 1 continued on next page*

*Figure 1 continued*

models that may explain the data and to test theories of social generalisation. Model M1 assumes participants are subject to both self-insertion and social contagion, i.e., participants used their own preferences as a prior to their partner in phase 2, and partner behaviour subsequently influenced participants' preferences in phase 3. Model M4 assumes participants are subject to neither self-insertion nor social contagion, instead forming a novel prior around the phase 2 partner rather than using their own preferences and failing to be influenced by their partner after observation. Models M2 and M3 suggest participants are only explained by either self-insertion or social contagion, not both. (**C**) We assume that participants' choices in phase 1 are governed by both a median ($\theta_{ppt}^{m}$) and standard deviation ($\theta_{ppt}^{\sigma}$). Participants insert their median preferences ($\theta_{ppt}^{m}$) into their prior beliefs over their partner in phase 2, but with a different standard deviation to allow for flexibility and learning ($\theta_{ref}^{\sigma}$). The combination of the prior and posterior belief uncertainty about the partner ($\theta_{ref}^{\sigma}$; $\theta_{par}^{\sigma}$), the precision participants have over their own preferences ($\theta_{ppt}^{\sigma}$), and the median posterior of the participant and partner ($\theta_{ppt}^{m}$; $\theta_{par}^{m}$) form the new median and standard deviation over participant preferences in phase 3 ($\theta_{ppt}$). (**D**) In contrast to M1, M4 generates a new central tendency over the partner in phase 2 ($\bar{\theta}_{par}^{m}$) which disconnects participant preferences and prior beliefs. M4 also assumes that the same parameters that generated participant choices in phase 1 also generate choices in phase 3. (**E**) Simulating our model demonstrates how different combinations of $\alpha$ (preferences for absolute self-reward) and $\beta$ (preferences for relative reward; prosocial-competitiveness) lead to changes in the discrepancy of value between participants and partners (left panel). We also show how increasing uncertainty over self-beliefs and higher precision over partners causally draws participants more toward the beliefs of their partner in phase 3 and increases their precision over their phase 3 beliefs (*Moutoussis et al., 2016*).

social value orientation paradigm – the Intentions Game (*Figure 1A*) – with virtual partners. In phase 1, participants made forced choices between two options for splitting points with an anonymous partner. In phase 2, participants learned to predict the decisions of a new anonymous partner using the same forced-choice set-up, receiving feedback on the accuracy of their successive predictions. Notably, using a novel server architecture (*Burgess and Barnby, 2023*), partners in phase 2 were configured to be approximately 50% different from the participants in terms of their choices, ensuring that all participants had to learn about their partners. Phase 3 mirrored the first, with participants informed that they were matched with a third anonymous partner, unrelated to those in phases 1 and 2. Detailed descriptions of the task can be found in the Materials and methods section and *Figure 1*.

**Table 1.** Demographics of participants.
CTQ = Childhood Trauma Questionnaire, MZQ = Mentalisation Questionnaire, RGPTSB = Revised Green Paranoid Thoughts Scale (Persecutory Subscale), RGPTSA = Revised Green Paranoid Thoughts Scale (Referential Subscale), CAMSQ = Certainty About Mental States Questionnaire. ETMCQ = Epistemic Trust, Mistrust, and Credulity Questionnaire, M=Male, F=Female, O=Other. For continuous variables, all means are stated with corresponding standard deviation in brackets. Significant differences are highlighted in bold.

| | BPD | CON | Test |
|---|---|---|---|
| **DEMOGRAPHICS** | | | |
| n | 50 | 53 | |
| Age | 31.2 [11.16] | 30.0 [8.64] | t=−0.61; p=0.54 |
| Gender (M:F:O) | 8:39:2 | 14:36:3 | t=−1.25; p=0.21 |
| Education (years) | 14.39 [3.38] | 14.8 [3.10] | t=0.62; p=0.53 |
| Soc. Dep. Index | 12834.94 [7911] | 11967.63 [7567] | t=−0.54; p=0.59 |
| **PSYCHOMETRICS** | | | |
| RGPTSB | 14.53 [1.07] | 5.33 [0.72] | **t=4.67; p<0.001** |
| RGPTSA | 16.79 [0.96] | 7.29 [0.70] | **t=6.47; p<0.001** |
| CTQ (trauma) | 64.88 [0.90] | 42.27 [0.78] | **t=6.48; p<0.001** |
| CAMSQ (self) | 4.95 [1.42] | 5.02 [0.22] | t=−0.07; p=0.94 |
| CAMSQ (other) | 5.32 [1.42] | 5.15 [0.17] | t=−0.71; p=0.48 |
| MZQ (mentalizing) | 55.94 [0.59] | 38.16 [0.85] | **t=9.39; p<0.001** |
| ETMCQ (Mistrust) | 5.25 [0.93] | 4.21 [0.84] | **t=5.27; p<0.001** |
| ETMCQ (Trust) | 4.86 [1.13] | 5.04 [0.86] | t=−0.87, p=0.39 |
| ETMCQ (Credulity) | 4.35 [1.06] | 3.34 [0.77] | **t=4.14; p<0.001** |

All participants also self-reported their trait paranoia, childhood trauma, trust beliefs, and trait mentalising (see Materials and methods).

## Psychometric and behavioural results

Participants with BPD, compared to CON, retrospectively reported significant childhood trauma, epistemic disruptions (including mistrust and credulity), elevated referential and persecutory beliefs, and demonstrated ineffective trait mentalising (*Table 1*). The groups did not differ in trait measures of certainty regarding self and others' mental states, nor in epistemic trust.

We analysed the 'types' of choices participants made in each phase (*Supplementary file 1*). The interpretation of a participant's choice depends on both values in a choice. For example, a participant could make prosocial (self = 5; other = 5) vs individualistic (self = 10; other = 5) choices, or prosocial (self = 10; other = 10) vs competitive (self = 10; other = 5) choices. There were 12 of each pair in phases 1 and 3 (individualistic vs prosocial; prosocial vs competitive; individualistic vs competitive).

In phase 1, both CON and BPD participants made prosocial over competitive choices with similar frequency (CON = 9.67 [3.62]; BPD = 9.60 [3.57]; $t=-0.11$, p=0.91). However, CON participants made significantly fewer prosocial choices when individualistic choices were available (CON = 2.87 [4.01]; BPD = 5.22 [4.54]; $t=2.75$, p=0.007). Both groups favoured individualistic over competitive choices with similar frequency (CON = 11.03 [1.95]; BPD = 10.34 [2.63]; $t=-1.52$, p=0.13). For a reaction time assessment, see *Supplementary file 5*.

Each group showed good predictive accuracy (CON = 77.2% [13.9%]; BPD = 72.7% [15.6%]). There was no difference in overall predictive accuracy between BPD and CON (linear estimate = 2.44, 95% CI: –0.67, 5.54; $t=1.56$; p=0.12), nor on a trial-by-trial basis (linear estimate = 0.26, 95% CI: –0.06, 0.59; $z=1.61$, p=0.11). All participants showed an effect of time on accuracy, such that participants became more accurate in predicting their partner over the course of phase 2 (linear estimate = 0.013, 95% CI: 0.008, 0.017; $z=6.01$; p<0.001). Server matching between participant and partner in phase 2 was successful, with participants being approximately 50% different to their partners with respect to the choices each would have made on each trial in phase 2 (mean similarity = 0.49, SD = 0.12).

In phase 3, both CON and BPD participants continued to make equally frequent prosocial vs competitive choices (CON = 9.15 [3.91]; BPD = 9.38 [3.31]; $t=-0.54$, p=0.59); CON participants continued to make significantly less prosocial vs individualistic choices (CON = 2.03 [3.45]; BPD = 3.78 [4.16]; $t=2.31$, p=0.02). Both groups chose equally frequent individualistic vs competitive choices (CON = 10.91 [2.40]; BPD = 10.18 [2.72]; $t=-0.49$, p=0.62).

## Computational analysis

Over all three phases, we assumed participants and their partners used a Fehr-Schmidt utility function (*Fehr and Schmidt, 1999*) to calculate the utility of two options $\left(\mathbf{U}_{\alpha,\beta} = \left\{\mathrm{U}^1_{\alpha,\beta},\ \mathrm{U}^2_{\alpha,\beta}\right\}\right)$, based on the joint rewards available for both the participant, $\mathbf{R}_{ppt} = \left\{\mathrm{r}^1_{ppt}, \mathrm{r}^2_{ppt}\right\}$, and their partner, $\mathbf{R}_{par} = \left\{\mathrm{r}^1_{par}, \mathrm{r}^2_{par}\right\}$. The utility of each option was weighted based on absolute-reward gain $\alpha$ (how much participants care about self-earnings, irrespective of the other) and relative-reward $\beta$ along a prosocial-competitive axis (how much participants care about equality of outcomes); $\beta < 0$ is prosocial, whereas $\beta > 0$ is competitive.

$$\mathbf{U}_{\alpha,\beta} = \alpha * \mathbf{R}_{ppt} + \beta * \max\left(\mathbf{R}_{ppt} - \mathbf{R}_{par},\ 0\right)$$

We then constructed four models (*Table 2*) to explain how participants used their own preferences ($\theta^m_{ppt} = \left\{\alpha^m_{ppt}, \beta^m_{ppt}\right\}$) and uncertainty over these preferences ($\theta^\sigma_{ppt} = \left\{\alpha^\sigma_{ppt}, \beta^\sigma_{ppt}\right\}$) to predict and learn about the preferences of their partner ($\theta_{par}$; *Figure 1B*; see Materials and methods). Model M1 (*Figure 1C*) suggests that participants initially use their own preferences as a prior belief about their partner (self-insertion), which is gradually diminished during the learning process in phase 2. M1 also posits that, following learning, the inferred beliefs about a partner will influence participants' own preferences, making them more similar to their partner's preferences following observation (social contagion). According to this model, participants shift towards their partner based on their uncertainty about self and others (*Figure 1E*): greater uncertainty over self-preferences and increased precision in representing the other cause stronger social contagion effects.

**Table 2.** Parameter and model specification.

Grey shading = parameters relevant to representations of the self (ppt). Orange shading = parameters relevant to representations of the other (par). Free = parameters are random variables to fit through model inversion. Derived = parameter is calculated from latent values within the model. SD = standard deviation.

| | M1 | M2 | M3 | M4 | Beta | Description | Type | Phase |
|---|---|---|---|---|---|---|---|---|
| $\alpha_{ppt}^m$ | X | X | X | X | | Median of absolute reward preferences | Free | 1 |
| $\beta_{ppt}^m$ | X | X | X | X | X | Median of relative reward preferences | Free | 1 |
| $\alpha_{ppt}^\sigma$ | X | X | X | X | | SD of absolute reward preferences | Free | 1 |
| $\beta_{ppt}^\sigma$ | X | X | X | X | X | SD of relative reward preferences | Free | 1 |
| $\underline{\alpha}_{par}^m$ | | | X | X | | Prior beliefs median over absolute reward preferences | Free | 2 |
| $\underline{\beta}_{par}^m$ | | | X | X | | Prior beliefs median over relative reward preferences | Free | 2 |
| $\alpha_{par}^{ref}$ | X | X | X | X | | Prior beliefs SD over absolute reward preferences | Free | 2 |
| $\beta_{par}^{ref}$ | X | X | X | X | X | Prior beliefs SD over relative reward preferences | Free | 2 |
| $\alpha_{par}^m$ | | X | X | X | | Posterior belief median over absolute reward preferences | Derived | 2 |
| $\beta_{par}^m$ | | X | X | X | X | Posterior belief median over relative reward preferences | Derived | 2 |
| $\alpha_{par}^\sigma$ | | X | X | X | | Posterior belief SD over absolute reward preferences | Derived | 2 |
| $\beta_{par}^\sigma$ | | X | X | X | X | Posterior belief SD over relative reward preferences | Derived | 2 |
| $\alpha_{ppt}^m$ | X | | X | | | Shifted median of absolute reward preferences | Derived | 3 |
| $\beta_{ppt}^m$ | X | | X | | X | Shifted median of relative reward preferences | Derived | 3 |
| $\alpha_{ppt}^\sigma$ | X | | X | | | Shifted SD of absolute reward preferences | Derived | 3 |
| $\beta_{ppt}^\sigma$ | X | | X | | X | Shifted SD of relative reward preferences | Derived | 3 |
| No. Free | 6 | 6 | 8 | 8 | 3 | | | |

Model M4 (*Figure 1D*), on the other hand, suggests that participants do not engage in these generalisation processes: predictions about others are not grounded in the self, and observing others does not alter self-preferences. Models M2 and M3 allow for either self-insertion or social contagion to occur independently. Consistent with prior research, we also constructed a model that assumes the same insertion and contagion processes as M1, but along a single prosocial-competitive axis ('Beta model'; *Barnby et al., 2022*). The 'Beta model' is equivalent to M1 in its architecture (both self-insertion and social contagion are hypothesised to occur) but differs in its utility function: participants might only consider a single dimension of relative reward allocation, which is typically emphasised in previous studies (e.g. *Hula et al., 2018*).

All computational models were fitted and compared using a hierarchical Bayesian inference (HBI) algorithm, which allows hierarchical parameter estimation while assuming random effects for group and individual model responsibility (*Piray et al., 2019*; see Materials and methods for more information). We report individual- and group-level model responsibility, in addition to exceedance probabilities between groups to assess model dominance.

## Model comparison – BPD participants hold disintegrated self-other beliefs

We found that CON participants were best fit at the group level by M1 (frequency = 0.59, exceedance probability = 0.98), whereas BPD participants were best fit by M4 (frequency = 0.54, exceedance probability = 0.86; *Figure 2A*). This suggests CON participants are best fit by a model that fully integrates self and other when learning, whereas those with BPD are best explained as holding disintegrated and separate representations of self and other that do not transfer information back and forth.

We first explore parameters between separate fits (see Materials and methods). Later, in order to assuage concerns about drawing inferences from different models, we examined the relationships between the relevant parameters when we forced all participants to be fit to each of the models (in a hierarchical manner, separated by group). In sum, our model comparison is supported by convergence in parameter values when comparisons are meaningful (see *Supplementary file 2*). We refer to both types of analysis below.

## Generative accuracy and recovery

We simulated data for each participant using their individual parameters from the winning model within each group and refitted our models using this simulated data. Model comparison yielded very similar results (*Figure 3A*): CON synthetic participants best fit at the group level by M1 (requency = 0.58, exceedance probability = 0.98) and BPD synthetic participants best fit by M4 (frequency = 0.57, exceedance probability = 0.85). The simulated data closely matched the actions of participants across all three phases (median accuracy = 0.8, SD = 0.12). In phase 2, the model-predicted total correct scores were not significantly different from observed scores (*Figure 3E*). Both model responsibility and common parameters within each dominant model were significantly associated (model confusion $\rho$=0.46–0.97, p<0.001; parameter recovery $\rho$=0.70–0.94, p<0.001; *Figure 3C*). $\beta_{ppt}^m$ preferences in phase 1 were negatively correlated with prosocial vs competitive choices (r=−0.77, p<0.001) and individualistic vs competitive choices (r=−0.59, p<0.001); $\alpha_{ppt}^m$ was positively correlated with individualistic vs competitive choices (r=0.53, p<0.001) and negatively correlated with prosocial vs individualistic choices (r=−0.69, p<0.001). Given the very good to excellent performance of the models, we analysed individual parameters and simulations between groups.

## Phase 1 – BPD participants are more certain about themselves

We first examined self-representations of participants in phase 1. CON participants (under model M1) and BPD participants (under M4) were equally prosocial (CON mean $[\beta_{ppt}^m]$=−7.50; BPD mean $[\beta_{ppt}^m]$=−6.59; $\Delta\mu\,[\beta_{ppt}^m]$ = 0.92, 95%HDI: −1.24, 3.12) – both groups valued equal allocation of reward between themselves and their partners. BPD participants had lower preferences for earning higher absolute rewards (CON mean $[\alpha_{ppt}^m]$=18.41; BPD mean $[\alpha_{ppt}^m]$=10.57; $\Delta\mu\,[\alpha_{ppt}^m]$ = −7.83, 95%HDI: −11.06, −4.75). BPD participants were also more certain about both self-preferences for absolute and relative reward ($\Delta\mu\,[\alpha_{ppt}^\sigma]$=−0.89, 95%HDI: −1.01, −0.75; $\Delta\mu\,[\beta_{ppt}^\sigma]$ = −0.32, 95%HDI: −0.60, −0.04) vs CON participants (*Figure 2B*).

These differences were replicated when considering parameters between groups when we fit all participants to the same models (M1-M4; see *Supplementary file 2*).

## Phase 2 – BPD participants use disintegrated and neutral priors

We next assessed how participants generated their prior beliefs about a partner in phase 2. CON participants were best fit by M1, which assumes the same median belief participants use in phase 1 is identical to their median prior belief about their partners. In contrast, BPD participants were best fit by M4 and so generated a new median prior belief about their partners. Assessing by individual models shows this was driven by expectations about a partner's prosocial-competitive preferences (relative reward; see *Supplementary file 2*).

Prior work predicts those with BPD should focus more intently on social information rather than private information that only concerns one party (*Henco et al., 2020*). In BPD participants, only new beliefs about the relative reward preferences – mutual outcomes for both players – of partners differed (see *Figure 2E*): new median priors were larger than median preferences in phase 1 (mean $[\beta_{par}^m]$=−0.47; $\Delta\mu\,[\beta_{ppt}^m - \beta_{par}^m]$ = −6.10, 95%HDI: −7.60, −4.60). BPD priors about their partner's relative preferences were also centred closely around 0 ($\Delta\mu\,[0 - \beta_{par}^m]$=−0.39, 95%HDI: −0.77, −0.05),

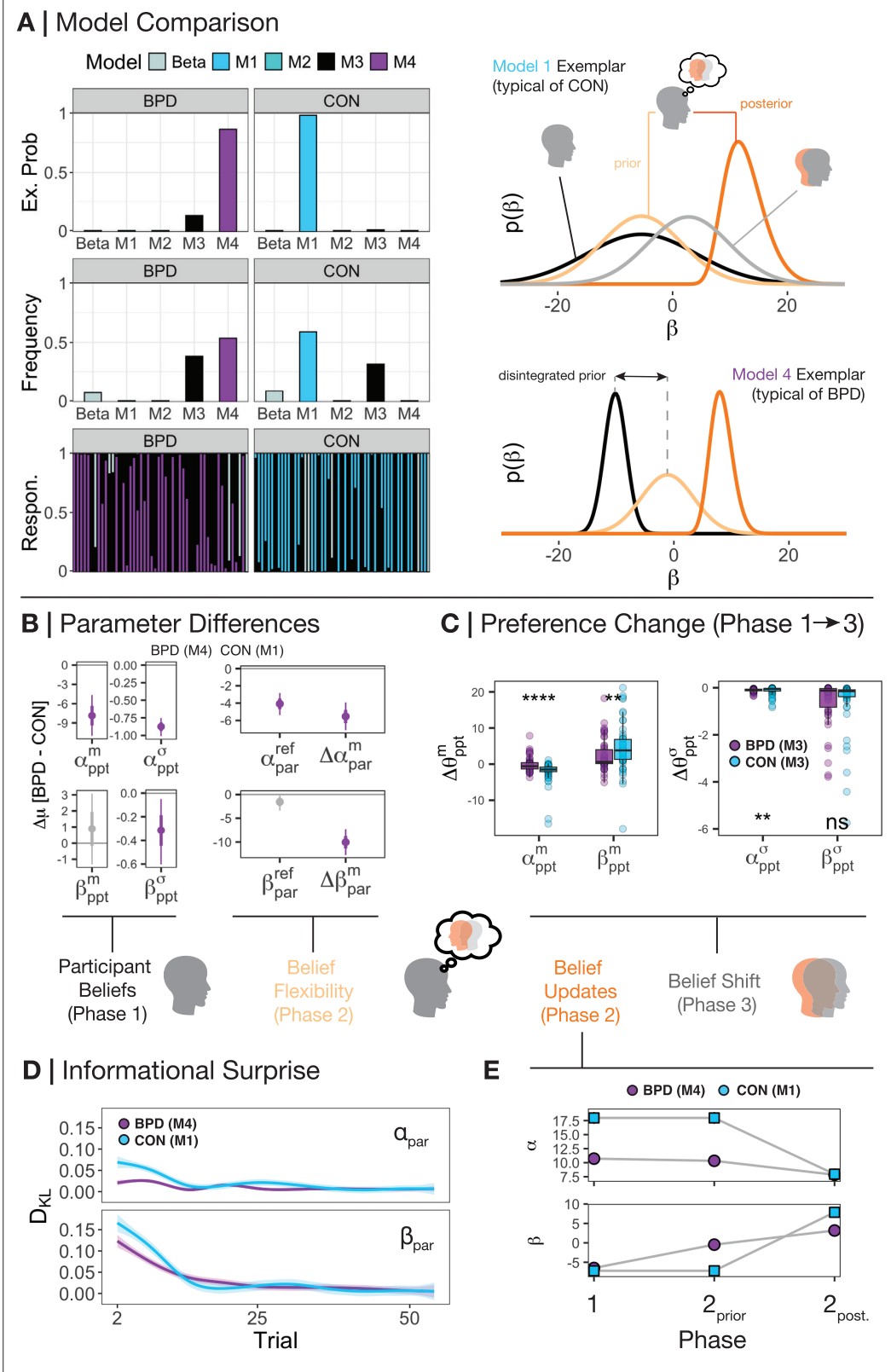

**Figure 2.** Beliefs between groups and within phases. (**A**) We used random-effects hierarchical model fitting and comparison to jointly estimate group-level and individual-level parameters based on real data from participants (**Piray and Daw, 2021**). CON participants were best fit by M1, whereas borderline personality disorder (BPD) participants were best fit by M4 on a group level. Looking within each model by simulating the beliefs of each

*Figure 2 continued on next page*

*Figure 2 continued*

participant reveals that – as expected – CON participants use the median of their self-preferences (black distribution) as a basis for their prior beliefs about partners (light orange distribution), and that the precision of their posterior beliefs about partners (dark orange distribution) and of their own self-preferences leads to a shifted model of the self (grey distribution). BPD participants, on the other hand, have a disintegrated prior over their partner which is not subject to their own self-representation. Likewise, there is no change in self-preferences following learning, and thus an absence of the light grey distribution. For illustration, we focus on beliefs over relative preferences ($\beta$) and use real individual participants as exemplars for illustration. (**B**) Across models, we extracted the common parameters that generate the behaviour of both CON and BPD participants – i.e., their median and standard deviation over both $\alpha$ (absolute reward preferences) and $\beta$ (relative value preferences), the flexibility over participants' prior beliefs about their partners over each dimension, and the absolute change in posterior beliefs in phase 2 over each dimension ($\Delta\alpha_{par}^{m}$; $\Delta\beta_{par}^{m}$). Using hierarchical Bayesian t-tests, we demonstrated the mean difference in parameter values between groups. Purple values lower than 0 indicate that the BPD participants had significantly smaller parameter values. Here, we find that BPD participants were less individualistic, equally prosocial, and more certain about their self-preferences. BPD participants were also less flexible over their beliefs about a partner's absolute reward preferences and updated their beliefs less across the board. (**C**) Examining participants under a blanket assumption that participants in both BPD and CON groups were influenced by their partner (model M3) revealed that BPD participants were significantly less influenced by their partner across the board, both with respect to their phase 3 median and standard deviation of beliefs. Kruskal-Wallis tests were used between groups within the visualisation. \*=p<0.05, \*\*=p<0.01, \*\*\*=p<0.001, \*\*\*\*=p<0.0001. (**D**) We also calculated the Kullback-Leibler divergence ($D_{KL}$) of beliefs between each trial ($t$-1 vs $t$) on each trial during phase 2. We observed three things: (1) All participants display larger belief updates initially, (2) all participants 'cool off' in their belief updating over the course of phase 2, and (3) BPD participants update beliefs significantly less throughout the course of phase 2 vs CON participants. (**E**) Examining the shift in central tendencies across both groups demonstrates that between phase 1 and the start of phase 2, BPD participants shifted their central tendency only over beliefs about the relative reward preferences of partners'. They held lower expectations about a partner's absolute reward preferences consistent with their own preferences.

The online version of this article includes the following figure supplement(s) for figure 2:

**Figure supplement 1.** Group-level parameter values.

**Figure supplement 2.** Individual-level parameter distributions per group.

**Figure supplement 3.** Group Distributions Across Phase 1 and 2.

**Figure supplement 4.** 2D distribution of participant and partner parameters estimated through Bayesian inference at the Amazon Web Service (AWS) server backend during the participant-partner matching protocol.

**Figure supplement 5.** Relationship between belief updates and reaction times.

**Figure supplement 6.** Uncorrected psychometric correlations.

suggesting that BPD participants entered into the interaction with very neutral priors about their partner's preferences for relative reward.

Models of moral preference learning (***Story et al., 2024b***) predict that BPD vs non-BPD participants have more rigid beliefs about their partners. We found that BPD participants were equally flexible around their prior beliefs about a partner's relative reward preferences ($\Delta\mu\left[\beta_{par}^{ref}\right]$=–1.60, 95%HDI: –3.42, 0.23), and were less flexible around their beliefs about a partner's absolute reward preferences ($\Delta\mu\left[\alpha_{par}^{ref}\right]$=–4.09, 95%HDI: –5.37, –2.80), vs CON (***Figure 2B***). Median belief change (from priors to posteriors) in phase 2 was lower in BPD vs CON ($\Delta\mu\left[\Delta\alpha_{par}^{m}\right]$=–5.53, 95%HDI: –7.20, –4.00; $\Delta\mu\left[\Delta\beta_{par}^{m}\right]$=–10.02, 95%HDI: –12.81, –7.30). Posterior beliefs about partner were more precise in BPD vs CON ($\Delta\mu\left[\alpha_{par}^{\sigma}\right]$=–0.94, 95%HDI: –1.50, –0.45; $\Delta\mu\left[\beta_{par}^{\sigma}\right]$ = –0.70, 95%HDI: –1.20, –0.25). This is unsurprising given the disintegrated priors of the BPD group in M4, meaning they need to 'travel less' in state space. Even under assumptions of M1-M4 for both groups, BPD vs CON showed smaller posterior standard deviation in phase 2 after learning (see ***Supplementary file 2***). These results converge to suggest those with BPD form rigid posterior beliefs.

We checked that conclusions about self-insertion did not depend on the different models. We found that $\beta_{par}^{ref}$ under M1 and M2 was consistently larger in BPD vs CON. This supports the notion that new central tendencies for BPD participants in phase 2 were required for expectations about a partner's relative reward (see ***Figure 3—figure supplement 2*** and ***Supplementary file 2***). $\alpha_{par}^{ref}$ and $\beta_{par}^{ref}$

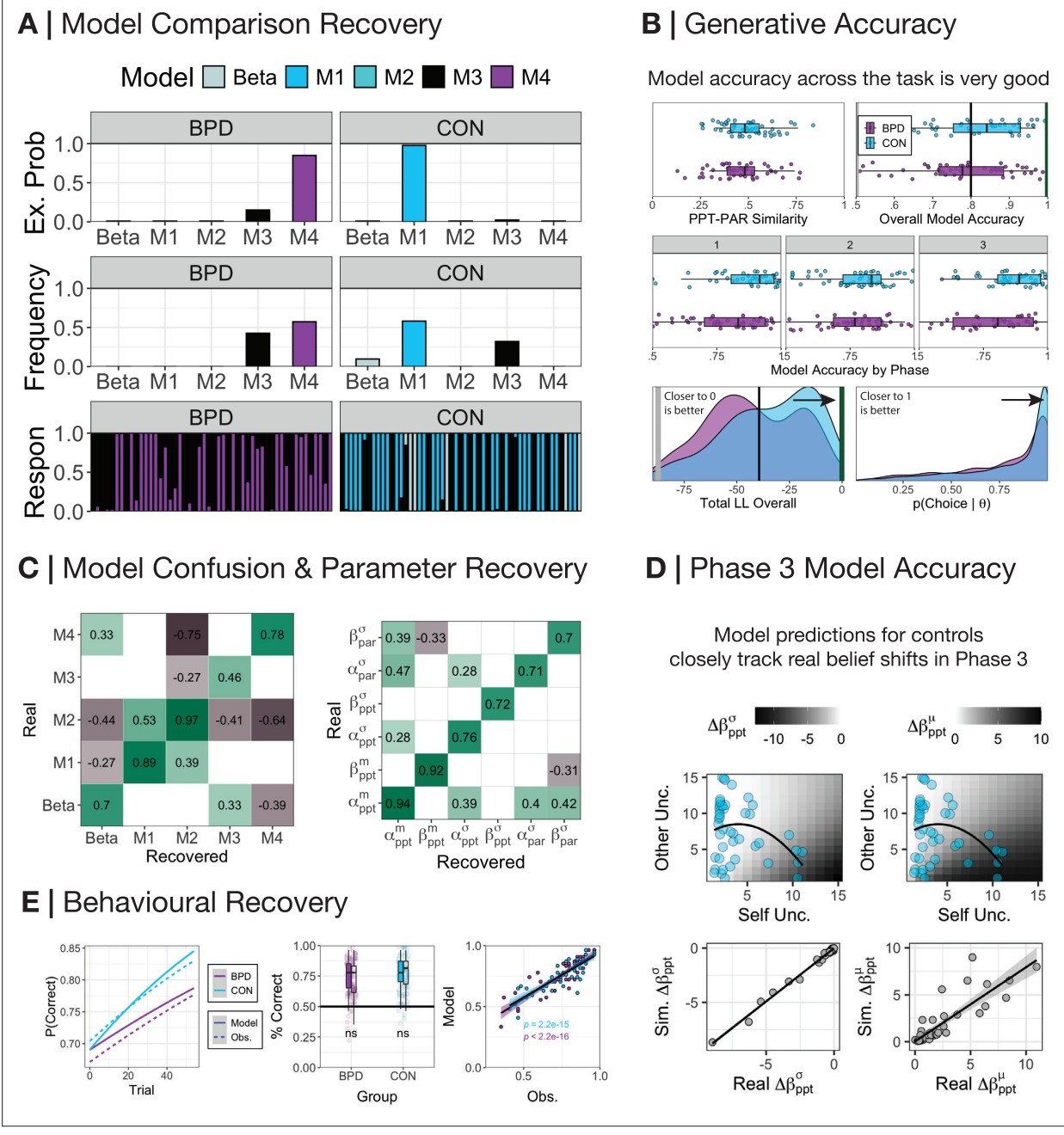

**Figure 3.** Model accuracy. (**A**) We used random-effects hierarchical model fitting and comparison to jointly estimate group-level and individual-level parameters on simulated data (*Piray et al., 2019*). CON participants were best fit by M1, whereas borderline personality disorder (BPD) participants were best fit by M4. (**B**) Server matching between participant and partner in phase 2 was successful, with participants being approximately 50% different to their partners with respect to the choices each would have made on each trial in phase 2 (mean similarity = 0.49, SD = 0.12). Model accuracy across the task was very high (mean accuracy = 0.8, SD = 0.12). Model accuracy within each phase was very high (mean accuracy[phase1]=0.83, SD[phase1]=0.16; mean accuracy[phase2]=0.77, SD[phase2]=0.14; mean accuracy[phase3]=0.82, SD[phase3]=0.17). Log-likelihood values were also well above what would be expected had the model fitted the data by chance (median = −40.68, SD = 22.7; chance value = −87.33). Choice probabilities generated by the model on each trial were also well above chance thresholds (median = 0.91, SD = 0.24; chance value = 0.5). (**C**) The Spearman association between the responsibility allocated for each participant during real and recovered model comparison was highly correlated on the diagonal. There was some correlation between M1 and M2, but this was due to M2 being a nested model of M1, sharing similar free parameters; this was not worrying in light of excellent model identifiability overall in the synthetic comparison. Associations between real and recovered parameters from the dominant model within each BPD and CON participants were very high with few cross correlations on the off-diagonal. In both confusion and parameter recovery matrices, white spaces indicate insignificant associations at the p>0.01 level. (**D**) (Top panel) The relationship between uncertainty over the self and uncertainty over the other with respect to the change in the precision (left) and median shift (right) in phase 3 relative reward

*Figure 3 continued on next page*

*Figure 3 continued*

preferences ($\beta_{ppt}$). CON participant self and other uncertainty is overlaid onto the plot to demonstrate the degree to which their beliefs *should* change in phase 3 according to the model. (Bottom panel) Correlating the model-predicted median shift in beliefs and derived change in beliefs between phases 1 and 3 demonstrates a very strong association ($r$=0.88, p<0.001). For the purposes of visualisation, we cap real and simulated values <15 for compactness, although the true correlation reported is irrelevant to this visual constraint. (**E**) (Left panel) We overlay model-predicted (solid line) and real observed (dashed line) trial-by-trial probabilities extracted from a linear model for a correct prediction by participants. For raw trial-by-trial updating, see *Figure 3—figure supplement 1*. Both closely match. (Middle panel) There was no significant difference (ns) for BPD and CON participants with respect to their total correct answers over phase 2. (Right panel) Model-predicted and real observations in phase 2 total scores were highly correlated in both groups (CON $r$=0.84, p<0.001; BPD $r$=0.89, p<0.001).

The online version of this article includes the following figure supplement(s) for figure 3:

**Figure supplement 1.** Generative Capacity of the Winning Model.

**Figure supplement 2.** Phase 2 prior belief flexibility following forced hierarchical fit of model M1 to all (FULL) participants and separate (SEP) groups.

**Figure supplement 3.** Pearson correlation between parameters of equivalence across models M1-M4.

**Figure supplement 4.** Simulation of phase 2 priors that may be drawn from a memory of an aversive other vs from the self alone.

parameters under assumptions of M1 and M2 were strongly correlated with median change in belief between phases 1 and 2 under M3 and M4, suggesting convergence in model outcome (*Figure 3— figure supplement 3*).

Analysing belief updating on a more granular trial-by-trial basis using M1 for CON and M4 for BPD revealed preference type and between-group differences in belief refinement over the course of phase 2 (*Figure 2D*). We examined this by analysing the Kullback-Leibler divergence ($D_{KL}$) – expected informational surprise – on each trial in Phase 2.

Across both groups and belief types, informational surprise reduced over time (linear estimate $D_{KL}$=–0.007, 95% CI: −0.008,−0.005; $t$=−7.60, p<0.001). Beliefs about a partner's relative reward preferences were updated more than absolute reward preferences (linear estimate = 0.54, 95% CI: 0.47, 0.62; $t$=14.00, p<0.001). These interacted, updating over relative vs absolute beliefs reduced over the course of phase 2 (linear estimate = –0.013, 95% CI: −0.015, −0.011; $t$=−10.81, p<0.001). These findings were supported under M1-M4-only assumptions (see *Supplementary file 3*).

BPD informational surprise is consistently restricted over beliefs about absolute reward vs CON. CON participants remained more flexible than BPD participants along both types of preference (linear estimate $[D_{KL}(\alpha_{par}^m)]$=0.40, 95% CI: 0.29, 0.51, $t$=7.18, p<0.001; linear estimate $[D_{KL}(\beta_{par}^m)]$=0.17, 95% CI: 0.29, 0.51, $t$=3.06, p=0.002). There was a group by time interaction, such that the difference between groups and the magnitude of belief updates decreased over time (linear estimate $[D_{KL}(\alpha_{par}^m)]$=–0.009, 95% CI:−0.012, –0.006, $t$=−5.30, p<0.001; linear estimate $[D_{KL}(\beta_{par}^m)]$=–0.004, 95% CI:−0.008, –0.001, $t$=−2.78, p=0.005) – CON participants and BPD participants eventually converged to an equivalent updating schedule (*Figure 2D*).

Assessing this same relationship under M1- and M2-only assumptions reveals a replication of this group effect for absolute reward, but the effect is reversed for relative reward (see *Supplementary file 3*). This accords with the context of each model, where under M1 and M2, BPD participants had larger phase 2 prior flexibility over relative reward (leading to larger initial surprise), which was better accounted for by a new central tendency under M4 during model comparison. When comparing both groups under M1-M4, informational surprise over absolute reward was consistently restricted in BPD (*Supplementary file 3*), suggesting a diminished weight of this preference when forming beliefs about an other.

We also explored how beliefs and choices were associated with reaction times, showing that belief updates and reaction times were coupled over the course of phase 2 and related to participant-partner similarity (*Figure 2—figure supplement 5*).

## Phase 3 – BPD participants are less influenced by partners

Prior work predicts that human economic preferences are shaped by observation (*Panizza et al., 2021*; *Suzuki et al., 2016*; *Yu et al., 2021*). Associative models also predict that social contagion may be exaggerated in BPD (*Story et al., 2024a*). In the dominant model for the BPD group – M4 – participants are not influenced in their phase 3 choices following exposure to their partner in phase 2. To further confirm this, we also analysed absolute change in median participant beliefs between phases 1

and 3 under the assumption that M1 and M3 were the dominant model for both groups (that allowed for contagion to occur). This analysis aligns with our primary model comparison using M1 for CON and M4 for BPD (*Figure 2C*). CON participants altered their median beliefs between phases 1 and 3 more than BPD participants (M1: linear estimate = 0.67, 95% CI: 0.16, 1.19; $t$=2.57, p=0.011; M3: linear estimate = 1.75, 95% CI: 0.73, 2.79; $t$=3.36, p<0.001). Relative reward was overall more susceptible to contagion vs absolute reward (M1: linear estimate = 1.40, 95% CI: 0.88, 1.92; $t$=5.34, p<0.001; M3: linear estimate = 2.60, 95% CI: 1.57, 3.63; $t$=4.98, p<0.001). There was an interaction between group and belief type under M3 (M3: linear estimate = 2.13, 95% CI: 0.09, 4.18, $t$=2.06, p=0.041) but not M1. There was a main effect of belief type on precision under M3 (linear estimate = 0.47, 95% CI: 0.07, 0.87, $t$=2.34, p=0.02) but not M1; relative reward preferences became more precise across the board. Derived model estimates of preference change between phase 1 and 3 strongly correlated between M1 and M3 along both belief types (see *Supplementary file 2* and *Figure 3—figure supplement 3*). As a whole, humans are more susceptible to changing relative preferences more than absolute reward preferences, and this is disrupted in BPD.

## Exploratory psychometric and intentional attribution analysis

Childhood trauma, persecution, and poor mentalising in BPD are all predicted to disrupt the integration of information from others (*Fonagy and Luyten, 2009*). Therefore, we explored whether social contagion may be restricted as a result of childhood trauma, paranoia, and less effective trait mentalising. We collected psychometric data from participants prior to entering the task and asked participants to attribute explicit intentions to their partner after phase 2. All analyses were corrected for false discovery rate (FDR; p[fdr]), and we provide correction for group status (*Supplementary file 4*).

We assessed conditional psychometric associations with social contagion under the assumption of M3 for all participants. We conducted partial correlation analyses to estimate relationships conditional on all other associations and retained all that survived bootstrapping (5000 reps), permutation testing (5000 reps), and subsequent FDR correction. When not controlled for group status, RGPTSB and CTQ scores were both moderately associated with MZQ scores (RGPTSB $r$=0.41, 95% CI: 0.23, 0.60, p[fdr]=0.043; CTQ $r$=0.354 95% CI: 0.13, 0.56, p[fdr]=0.02). This was not affected by group correction. CTQ scores were moderately and negatively associated with shifts in individualistic reward preferences ($\Delta\alpha_{ppt}^{m}$; $r$=–0.25, 95% CI: –0.46,–0.04, p[fdr]=0.03). This was not affected by group correction. MZQ scores were in turn moderately and negatively associated with shifts in prosocial-competitive preferences ($\Delta\beta_{ppt}^{m}$) between phases 1 and 3 ($r$=–0.26, 95% CI: –0.46, –0.06, p[fdr]=0.03). This was diminished when controlled for group status ($r$=0.13, 95% CI: –0.34, 0.08, p[fdr]=0.20). Together, this provides some evidence that self-reported trauma and self-reported mentalising are associated with social contagion (*Figure 2—figure supplement 6*). Social contagion under M3 was highly correlated with contagion under M1, demonstrating parsimony of outcomes across models (*Figure 3—figure supplement 3*).

Prior work has predicted that partner-participant preference disparity influences mental state attributions (*Barnby et al., 2022*; *Panizza et al., 2021*). We tested parameter influences on explicit intentional attributions in phase 2 while controlling for group status. Attributions included the degree to which they believed their partner was motivated by harmful intent (HI) and self-interest (SI). According to prior work (*Barnby et al., 2022*), greater disparity of absolute preferences before learning was associated on a trend level with reduced attributions of SI ($\rho\,[|\alpha_{par}^{m} - \alpha_{ppt}^{m}|]$=–0.23, p[fdr]=0.08), and greater disparity of relative preferences before learning exaggerated attributions of HI ($\rho\,[|\beta_{par}^{m} - \beta_{ppt}^{m}|]$=0.21, p[fdr]=0.08), but did not survive correction. This is likely due to partners being significantly less individualistic and prosocial on average compared to participants ($\Delta\mu\,[\alpha]$=–5.50, 95%HDI: –7.60, –3.60; $\Delta\mu\,[\beta]$ = 12, 95%HDI: 9.70, 14.00); partners are recognised as less selfish and more competitive.

Uncertainty in one's expectations is associated with subsequent mental state attributions of intentional harm (*Barnby et al., 2022*). Greater prior uncertainty (before interaction) over a partner's relative preferences was associated with increased HI ($\rho\,[\beta_{par}^{ref}]$=0.25, p[fdr]=0.04) but not SI, corrected for group status. There was no uncorrected association between prior uncertainty over absolute preferences with either attribution. Thus, expectations of greater difference between self and others may exaggerate beliefs about the intentional harm of others.

## Discussion

We built and tested a theory of interpersonal generalisation in a population of matched participants with (BPD) and without (CON) a diagnosis of BPD using the Intentions Game, a three-phase social value orientation task. We compared four hypotheses, instantiated in formal computational models, to determine whether those with a diagnosis of BPD displayed disrupted self-insertion and social contagion. Both groups demonstrated equivalent behavioural accuracy but employed different strategies. CON participants used a process of self-other generalisation to predict and align with their partners, while BPD participants maintained distinct representations of self and other, particularly over joint reward outcomes. As a whole, all participants were more sensitive to updates about joint vs absolute outcomes, with BPD participants particularly concerned with how outcomes relatively affected self and other. Our exploratory findings also indicate that retrospectively reported childhood trauma and persecutory beliefs were linked to reduced trait mentalising, which was subsequently associated with diminished shifts in participants' relative reward preferences. Collectively, our results integrate prior findings in BPD and provide a formal account of social information generalisation in humans, alongside a concise social paradigm to test these processes.

The data replicate models of social generalisation that have focused on individual processes of self-insertion and contagion, extending these theories by demonstrating both processes in conjunction. Models of self-insertion directly map participant preferences onto prior beliefs about others, which has been used to explain increased reaction times in observational learning of others' snack food preferences (*Tarantola et al., 2017*), as well as improved predictive accuracy when matched with individuals of similar social values (*Barnby et al., 2022*). Both findings are replicated in this study. Although we did not explicitly model reaction times, we observed an interaction between reaction time reductions over time and interpersonal similarity at baseline. In tandem, computational models of social contagion have focused on intertemporal discounting (*Moutoussis et al., 2016*) – with behavioural studies also focusing on effort-based reward (*Devaine and Daunizeau, 2017*) and moral preferences (*Yu et al., 2021*) – and explain shifts in self-preferences as a function of uncertainty regarding self and others. In both the dominant (M1) and sub-dominant (M3) models that best explained data in healthy participants, shifts in self-beliefs were also influenced by representational uncertainty of self and other: greater self-uncertainty and reduced other uncertainty led to larger shifts in social preferences.

The data also align with prior research on social impression formation, which suggests that humans form rapid evaluations of others that are refined over time (*Bone et al., 2021*; *Moutoussis et al., 2024*). This initial 'heating' and subsequent 'cooling' of beliefs corresponds to the computational complexity employed: model-based strategies are typically used early in interactions, transitioning to simpler, model-free computations once a partner's behaviour becomes predictable (*Gęsiarz and Crockett, 2015*; *Guennouni and Speekenbrink, 2022*). Our findings support this framework, demonstrating initial variability early in interactions followed by steady updating.

Disruptions in self-to-other generalisation provide an explanation for previous computational findings related to task-based mentalising in BPD. Studies tracking observational mentalising reveal that individuals with BPD, compared to those without, place greater emphasis on social over internal reward cues when learning (*Henco et al., 2020*; *Fineberg et al., 2018*). Those with BPD have been shown to exhibit reduced belief adaptation (*Siegel et al., 2020*) along with 'splitting' of latent social representations (*Story et al., 2024b*). BPD is also shown to be associated with overgeneralisation in self-to-other belief updates about individual outcomes when using a one-sided reward structure (where participant responses had no bearing on outcomes for the partner; *Story et al., 2024a*). Our analyses show that those with BPD are equal to controls in their generalisation of absolute reward (outcomes that only affect one player) but disintegrate beliefs about relative reward (outcomes that affect both players) through adoption of a new, neutral belief. We interpret this together in two ways: (1) There is a strong concern about social relativity when those with BPD form beliefs about others. (2) The absence of self-insertion when predicting relative outcomes may predispose to brittle or 'split' beliefs. In other words, those with BPD assume ambiguity about the social relativity preferences of another (i.e. how prosocial or punitive) and are quicker to settle on an explanation to resolve this. Although self-insertion may be counter-intuitive to rational belief formation, it has important implications for sustaining adaptive, trusting social bonds via information moderation.

Those with a diagnosis of BPD also show reduced permeability in other-to-self generalising. While prior research has predominantly focused on how those with BPD use information to form

impressions, it has not typically been examined whether these impressions affect the self. In interactive trust paradigms, neural responses to monetary offers from others to the self were substantially blunted in individuals with BPD compared to those without (**King-Casas et al., 2008**). Similarly, in non-social reward tasks, those with BPD show reduced neural feedback-related negativity amplitudes, which obstructs feedback-related self-change (**Stewart et al., 2019**; **Vega et al., 2013**). Our results suggest a mechanistic basis for social contagion, indicating that self-rigidity prevents observed social behaviours from generalising to the self, potentially exacerbated by childhood trauma, paranoia, and impaired mentalising capabilities. Resistance to social influence may serve as a protective response but can also contribute to the pervasive loneliness experienced by individuals with BPD, even in the absence of social isolation (**Liebke et al., 2017**).

Notably, despite differing strategies, those with BPD achieved similar accuracy to CON participants. While all participants were more concerned with relative vs absolute reward, those with BPD changed their strategy contingent on this focus. Practically, this difference in BPD is captured either through disintegrated priors with a new median (M4) or very noisy, but integrated priors over partners (M1) if we assume M1 can account for the full population. In either case, the algorithm underlying the computational goal for BPD participants is far higher in uncertainty, whether through a neutral central tendency (M4) or large variance (M1) prior over relative reward in phase 2, and emphasises a less stable or reliable expectation about others. It is important to assess this mechanism alongside momentary assessments of mood to understand whether more entropic learning processes contribute to distressing mood fluctuation.

Clinical implications of our work underscore the importance of consistency and stability in clinical support for individuals with a diagnosis of BPD. Encouragingly, we found that those with BPD were not entirely impermeable to observed behaviour, suggesting that consistent external models of trust could be internalised over time. Restoring a stable sense of self through social learning and effective mentalising (**Nolte et al., 2023**), along with a consistent focus on differentiating self from other (**Lowyck and Luyten, 2021**), is central to mentalisation-based therapies (**Bateman and Fonagy, 2010**; **Smits et al., 2024**) and other evidence-based treatments for BPD. We hope that our paradigm and model can offer insights into the effectiveness of these and other therapies in driving mechanistic psychological change. A key task for future work will be to assess whether generalisation principles may be restored in within-individuals with a diagnosis of BPD following intervention.

More broadly, our model bridges formal theories of associative learning and social cognition. Reinforcement learning approaches have effectively organised theories around uncertainty navigation in non-social contexts (**Piray and Daw, 2021**; **Zika, 2023**). However, humans do not function in isolation. Bayesian models of internal and external social beliefs are better suited to capture the dynamic nature of time, context, and uncertainty during interactions (**FeldmanHall and Nassar, 2021**; **Vélez and Gweon, 2021**), where joint reward rather than individual reward may be particularly salient (**Barnby et al., 2023**). Our paradigm is concise, visually engaging, includes straightforward rules and instructions, and allows for tight experimental control over partner similarity. Our model and paradigm effectively capture core social psychological principles grounded in general computational approaches to learning and uncertainty, elucidating key aspects of human social interaction and exchange.

We note some limitations to our study. Primarily, we focused on the ability of individuals to integrate their self-concept into beliefs about others. It is also possible that humans possess strong, salient representations of others (or groups of others) that serve as dominant templates for learning. This may be particularly relevant for individuals with BPD, who will often have interpersonal experiences of abuse, neglect, or other forms of distress. The use of a salient, negative other-prior as a basis for learning was not measured in this study, but it may explain the ambivalent prior observed in phase 2, where a mixture of self and notional other influences belief formation, leading to rigid belief updating. Individuals with BPD may integrate priors from different sources as a mixture. We can simulate this by modelling a framework that incorporates priors based on both self and a strong memory impression of a notional other (**Figure 3—figure supplement 4**). However, a strength of our data is that we observed impression formation independent of valence – impressions were formed regardless of whether a partner was more or less prosocial or selfish than the participant (**Figure 2—figure supplement 3**). This supports our hypothesis that a vulnerable self-model and lack of self-insertion contribute to the formation of overly precise beliefs during learning as a means of rapidly reducing uncertainty. Even if a mixture model better explains the ambivalent prior in

phase 2, it would still support a general hypothesis about the fractured concept of self and other in BPD.

Another strength of our work is demonstrating processes of self-insertion and contagion under minimal interaction conditions: simple observation alone was sufficient to elicit both processes. However, this is also a limitation. While we predict that these processes will apply in more naturalistic settings, this has yet to be tested, and it remains unclear whether these effects will persist in richer conditions, particularly when higher affective arousal and challenges to mentalising are present. Lastly, the action space and parameters governing choice in our study were quite simple – two actions influenced by two parameters. This was a deliberate computational choice to avoid overly complex action spaces that may be difficult to fit to real human data, and which might fail to capture how these mechanisms operate in the context of increasing action and model complexity. As a whole, our findings open new possibilities for testing how social uncertainty across the lifespan (e.g. in adolescence; *Sebastian et al., 2008*), and in the context of ill health, may explain the formation and maintenance of healthy social bonds, as well as their disruption.

Finally, a limitation may be that behaviour in tasks based on economic preferences may not have clinical validity. This issue is central to the field of computational psychiatry, much of which is based on generalising from tasks like that within this paper and discussing correlations with psychometric measures. Extrapolating economic tasks into the real world has been the topic of discussion for the many reviews on computational psychiatry (e.g. *Montague et al., 2012*; *Hitchcock et al., 2022*; *Huys et al., 2016*). We note a strength of this work is the use of model comparison to understand algorithmic differences between those with BPD and matched healthy controls. Nevertheless, we wish to further pursue how latent characteristics captured in our models may directly relate to real-world affective change.

## Materials and methods

### Data and code availability

All anonymised data, code, and custom software reported in this work are available on an open source licence: https://github.com/josephmbarnby/SocialTransfer_Barnby_etal_2024 copy archived at *Barnby, 2025*.

### Participants

We used a case-control, between-subjects design with 103 participants: a control group from the general population (*N*=53) and a clinical group diagnosed with BPD (*N*=50). Both groups were recruited for a larger study investigating social exchanges in BPD and anti-social personality disorder (approved by the Research Ethics Committee for Wales, 12/WA/0283; informed consent and consent to publish was obtained). The control and clinical groups were matched on age, sex, years in education, and the English Indices of Deprivation based on the 2019 census (IoD2019; Ministry of Housing, Communities & Local Government, 2019). Participants received £70 compensation for completing questionnaires and online tasks which included the Intentions Game. They also received a performance bonus if they were entered into the lottery for surpassing 1000 points over the course of the game.

Participants for the control group were recruited through an advertisement on the Call For Participants website (https://www.callforparticipants.com), local community services, and adult schools. Inclusion criteria required control participants to have no pre-existing or current diagnoses of mental health disorders, neurological disorders, or traumatic brain injuries. Additionally, control participants must not have been currently in therapy or taking medication for any psychiatric disorders.

The majority of BPD participants were recruited through referrals by psychiatrists, psychotherapists, and trainee clinical psychologists within personality disorder services across nine NHS Foundation Trusts in London and three NHS Foundation Trusts across England (Devon, Merseyside, Cambridgeshire). Four BPD participants were also recruited by self-referral through the UCLH website, where the study was advertised. To be included in the study, all participants needed to have, or meet criteria for, a primary diagnosis of BPD (or emotionally unstable personality disorder or complex emotional needs) based on a professional clinical assessment conducted by the referring NHS trust (for self-referrals, the presence of a recent diagnosis was ascertained through thorough discussion

with the participant, whereby two of the four also provided clinical notes). The patient participants also had to be under the care of the referring trust or have a general practitioner whose details they were willing to provide. Individuals with psychotic or mood disorders, recent acute psychotic episodes, severe learning disability, or current or past neurological disorders were not eligible for participation and were therefore not referred by the clinical trusts.

## Psychometric measures

### Green et al. Paranoid Thought Scale

The Green et al. Paranoid Thought Scale (GPTS) assesses paranoid thoughts, including ideas of social reference (scale A) and persecution (scale B), in both general and clinical populations (*Green et al., 2008*). Each item is scored from 0 (not at all) to 5 (totally) concerning endorsement of each item. We retained items from the GPTS that were consistent with the revised version outlined in *Freeman et al., 2021* (Revised GPTS; R-GPTS). The R-GPTS has demonstrated excellent psychometric properties (*Freeman et al., 2021*), making it a reliable and valid tool for assessing trait paranoid thoughts in non-clinical and clinical populations.

### Childhood Trauma Questionnaire

The Childhood Trauma Questionnaire (CTQ) is used to screen for maltreatment history (*Bernstein et al., 2003*). Each item is scored from 1 (never true) to 5 (very often true). The CTQ has shown good internal consistency reliability across the five scales (*Sacchi et al., 2018*) and good construct validity based on significant associations with stress responsivity (*McMahon et al., 2024*) and dissociation (*Nobakht et al., 2021*).

### Certainty About Mental States Questionnaire

The Certainty About Mental States Questionnaire (CAMSQ) assesses one's certainty in classifying the mental states of oneself and others at an abstract level (*Müller et al., 2023*), e.g., 'I know what other people think of me' and 'I know my feelings'. Each subscale is scored from 1 (never) to 7 (always). In US and German samples, the CAMSQ showed high internal consistency for Self-Certainty ($\omega$=0.90/0.88) and Other-Certainty ($\omega$=0.91/0.89) subscales, and high 2-week test-retest reliability for Self-Certainty ($r$=0.85), Other-Certainty ($r$=0.78), and Other-Self-Discrepancy ($r$=0.82) scores (*Müller et al., 2023*).

### Mentalisation Questionnaire

The Mentalisation Questionnaire (MZQ) is a 15-item questionnaire assessing an individual's trait mentalising, i.e., one's ability to understand and interpret their own and others' mental states (*Hausberg et al., 2012*). The MZQ demonstrated good internal consistency ($\alpha$=0.81) and test-retest reliability ($r$=0.76), and was sensitive to change over a 6-month follow-up period and showed good criterion-related validity, distinguishing individuals with BPD from those without BPD (*Hausberg et al., 2012*). A higher score reflects worse trait mentalising.

### Epistemic Trust, Mistrust, and Credulity Questionnaire

The Epistemic Trust, Mistrust, and Credulity Questionnaire (ETMCQ) is a 15-item measure calibrated to assess trust (e.g. 'I usually ask people for advice when I have a personal problem), mistrust (e.g. 'I'd prefer to find things out for myself on the internet rather than asking people for information), and credulity (e.g. 'I am often considered naïve because I believe almost anything that people tell me'; *Campbell et al., 2021*). Each item is scored from 1 (Strongly Disagree) to 7 (Strongly Agree).

## Paradigm, procedure, and server architecture

The Intentions Game is a repeated social-value orientation paradigm with three phases.

In phase 1 of the Intentions Game, participants take on the role of the decider with an anonymous partner over 36 trials. In each trial, participants choose between two options to distribute points between themselves and their partners. Participants make 12 choices each between prosocial and competitive (e.g. Option 1=[10,10], Option 2 = [10,5]) individualistic and competitive (e.g. Option 1=[10,5], Option 2=[8,1]), and prosocial and individualistic options (e.g. Option 1=[5,5], Option 2=[10,5]). Phase 1 choices allowed experimenters to classify participants' social preferences as

prosocial (preferring equal outcomes), individualistic (maximising own payoff), or competitive (maximising relative payoff difference at the cost of lower self-gain).

We included a task environment that balanced each type of choice pair (see *Supplementary file 1*).

In phase 2 of the game, participants were matched with a new anonymous partner and played the role of the recipient over 54 trials. In this phase, the participants predicted which of the two options their partner would choose on each trial. Trial numerical values for self and other were identical to phase 1. Partners' decisions were determined via a dynamic algorithm (*Burgess and Barnby, 2023*) to ensure partners were approximately ~50% different from the participants' based on participants' choices in phase 1. To surmise this architecture, we implemented a version of the client-server paradigm hosted on an Amazon Web Service (AWS) LightSail server, where the web-based behavioural task (implemented with JavaScript in Gorilla.sc) acted as the client and exchanged information with a remote AWS server. The server received all anonymised behavioural data following phase 1. The Application Programming Interface (API) to interact with the server used a customisable R script (v4.3) to process the received data from the participant, and additional R scripts were used to process and generate output for the participant. A function within the backend scripts first used Bayesian inference to approximate a participant's parameters for phase 1. It then simulated what choices the participant would have made in phase 2 had the participant been in the role of the partner. The algorithm then sought to find parameters that would be at least 50% dissimilar from participant parameters with respect to the generated choices of those parameters. This allowed the task behaviour of phase 2 to be dynamically updated in response to participant choices in phase 1. This facilitated tight control over the state of the task and enabled advanced computations to be performed on participant data beyond the capabilities of a web browser.

Participants were incentivised in phase 2 to predict accurately, as accurate predictions would contribute to their total point scores (total correct answers were multiplied by 10 and added to their points) and determined their entry into the lottery to win an extra £20 Amazon voucher. After participants had made their predictions, they were given feedback informed on whether their predictions were accurate.

At the end of phase 2, participants were asked to rate (1) the extent to which they thought their partner was driven by the desire to earn points in this task overall (self-interest) and (2) the extent to which they thought their partner was driven by the desire to reduce the participant's points in this task overall (attribution of harmful intent). The answers were presented using two separate sliders from 0 to 100; the sliders were initialised to be invisible until the participants made the first click.

Phase 3 was identical to phase 1 except that participants were matched with a new anonymous partner. Participants would take on the decider role similar to phase 1, which allowed experimenters to estimate whether the observation of their partner in phase 2 had an influence on participants in phase 3.

## Behavioural analysis

All analysis was conducted in R (v. 4.3.3) on a MacBook Pro (M2 Max; OS = Ventura13.5). All individual numeric values extraneous of statistical tests are reported with their mean and standard deviation (mean = XX, SD = YY). All statistical tests where dependent variables mapped one value to one participant (e.g. trait psychometric scores) were conducted as linear models, with the regression coefficient, 95% confidence interval (95% CI), t-value and p-value reported like so (linear estimate = XX, 95% CI:AA,BB; t=CC, p=DD). When dependent variables mapped multiple values to each participant (e.g. trial-by-trial accuracy or reaction time), random-effects linear modelling was used. All correlations used Pearson estimates ($r$) unless distributions were non-normal, in which case Spearman-ranked correlations ($\rho$) were performed.

## Model space and computational analysis

We apply four computational hypotheses (M1-M4) which could explain the data collected from the Intentions Game (*Figure 1*), centred around formal principles of self-insertion and social contagion. Self-insertion states that a self inserts their own preferences into their beliefs about others (*Andersen and Chen, 2002*; *Krueger and Clement, 1994*); social contagion states that a self's preferences will change when exposed to the preferences of another (*Deutsch and Gerard, 1955*). In each case,

cognitive representations of self and other are allowed to intermingle to form a new hybrid of the two for the purposes of computational efficiency and/or social bonding.

We note some important assumptions in our notation going forward. In dyadic social interaction, both parties are trying to estimate and predict the true state ($\theta$) of the self ($\theta_s$) and the other ($\theta_o$). However, this estimation is inherently imperfect. Theories of social inference need to consider three sources of noisy estimation of this quantity: the self's (s) metacognitive model of their own state, $\bar{\theta}_s$, their partner's (o) state, $\bar{\theta}_o$, and finally the experimenter's approximation of both quantities, $\theta_{s,o}$ (***Barnby et al., 2024***). In this work, we consider the experimenter's approximation of the self's state $\theta_s$ (phase 1), the self's approximation of their other $\bar{\theta}_o$ (phase 2), and how exposure to a partner may influence $\theta_s$ (phase 3). We term the self the participant (ppt) and the other the partner (par) and assume $\theta_{ppt} \equiv \hat{\theta}_s$ in phase 1, $\theta_{par} \equiv \bar{\theta}_o$ in phase 2, and $\theta_{ppt}$ are the shifted participant preferences following exposure to the partner.

All models assumed a constricted Fehr-Schmidt utility function was used by participants and partners to calculate the utility of two options $\left( \mathbf{U}_{\alpha,\beta} = \left\{ \mathrm{U}^1_{\alpha,\beta}, \mathrm{U}^2_{\alpha,\beta} \right\} \right)$ in each trial within the task.

In phase 1, participants made binary choices $c^t$, $t = \{1 \dots T\}$ about whether option 1 or option 2 should be chosen given the returns for each option pair, $\mathbf{R}^t = \left\{ \mathbf{R}^{t;1}; \mathbf{R}^{t;2} \right\} = \left\{ R^{t;1}_{ppt}, R^{t;1}_{par}, R^{t;2}_{ppt}, R^{t;2}_{par} \right\}$.

$$
\begin{aligned}
\mathbf{U}_{\alpha,\beta}\left(\mathbf{R}^{t;1}\right) &= \alpha_{\mathrm{ppt}} \cdot R^{t;1}_{\mathrm{ppt}} + \beta_{\mathrm{ppt}} \cdot \max\left(R^{t;1}_{\mathrm{ppt}} - R^{t;1}_{\mathrm{par}}, 0\right) \\
\Delta\mathbf{U}_{\alpha,\beta}\left(\mathbf{R}^t\right) &= \mathbf{U}_{\alpha,\beta}\left(\mathbf{R}^{t;1}\right) - \mathbf{U}_{\alpha,\beta}\left(\mathbf{R}^{t;2}\right)
\end{aligned}
\tag{1}
$$

Here, $\alpha_{ppt}$ describes the weight a participant places on their own payoff (in one reduced model, we set $\alpha_{ppt} = 1$), and $\beta_{ppt}$, the weight a participant places on their payoff relative to the payoff of their partner. Large positive or negative values of $\beta_{ppt}$ indicate, respectively, that participants like or dislike earning more than their partner. We can therefore describe these terms $\alpha$ and $\beta$ as reflecting preferences for absolute and relative payoffs, respectively. For efficiency, we discretised states of $\alpha_{ppt}$ from 0 to 30 (increments of 0.125) and $\beta_{ppt}$ from –30 to 30 (increments of 0.25).

Over this state space, we can construct a belief that participants are estimated to hold which generates their choices, **C**. Herein, we refer to this belief as $\theta_{ppt}$, where $\theta_{ppt}$ is a matrix over a fixed grid of $\alpha_{ppt}$ and $\beta_{ppt}$ values. In the models, $\theta_{ppt}$ is drawn from a normal distribution made from a central tendency, $\theta^m_{ppt}$, and a standard deviation, $\theta^\sigma_{ppt}$. The standard deviation around the central tendency allows for stochastic choice behaviour consistent with random utility models (***Block, 1974***; ***McFadden, 1974***). We invert the model to estimate $\theta_{ppt}$ based on a participant's choices given their likelihood of choosing $c^t = 1$ :

$$
\begin{aligned}
p\left(\theta_{\mathrm{ppt}} \mid \boldsymbol{C}\right) &\sim \mathcal{N}\left(\theta_{\mathrm{ppt}}; \theta^m_{\mathrm{ppt}}, \theta^\sigma_{\mathrm{ppt}}\right) \\
p\left(c^t = 1 \mid \theta_{\mathrm{ppt}}, \mathbf{R}^t\right) &= \sum_{\theta_{\mathrm{ppt}}} \sigma\left(\Delta U_{\alpha,\beta}\left(\mathbf{R}^t\right)\right) \cdot \theta_{\mathrm{ppt}} \\
LL &= \log\left[p\left(c^t = 1 \mid \theta_{\mathrm{ppt}}, \mathbf{R}^t\right)\right]
\end{aligned}
\tag{2}
$$

When $\theta^\sigma_{ppt}$ is larger, a participant's choices in phase 1 are estimated to be less deterministic and more stochastic – i.e., they are less sure about their preferences along each dimension. This consideration will become important for choices made in phase 3.

In phase 2, over 54 trials, we then model the participants' binary predictions $\bar{d}^t$, $t = \{1 \dots T\}$ about whether option 1 or 2 would be chosen by their partner given the returns $\mathbf{R}^t = \left\{ \mathbf{R}^{t;1}; \mathbf{R}^{t;2} \right\} = \left\{ R^{t;1}_{ppt}, R^{t;1}_{par}, R^{t;2}_{ppt}, R^{t;2}_{par} \right\}$ for each pair of options. They then were given feedback about the partner's true decision, which we note as $d^t$. We assumed the participant predicted the partner in the same way they would themselves, ranging along two dimensions, $\alpha_{par}$ and $\beta_{par}$, which was needed to be inferred through observation, using a likelihood for $d^t$ of $LL = \log\left[p\left(d^t = 1 | \alpha_{par}, \beta_{par}, \mathbf{R}^t\right)\right]$ using the same formula as phase 1. We note the belief about $\alpha_{par}$ and $\beta_{par}$ together as $\theta_{par}$, represented as a matrix over a fixed grid of $\alpha_{par}$ and $\beta_{par}$ values.

The partner decisions $D^t = \left\{ d^1, d^2 \dots, d^T \right\}$ are then used to update the participants' beliefs about the partner, written as $p\left(\theta_{par} | D^t\right)$, starting with prior $p\left(\theta_{par} | D^0\right)$. Both M1 and M2 assume participants use their own central tendency, $\theta^m_{ppt}$, as a starting point for their prior beliefs about their partner as theoretically outlined as a self-insertion bias (***Barnby et al., 2024***), which draws from past computational work (***Barnby et al., 2022***; ***Tarantola et al., 2017***). We also assumed participants used a new

standard deviation $\theta_{par}^{ref}$, which allowed for participants to believe their partner may be different from them (belief flexibility). Therefore, we have:

$$p\left(\theta_{par}|D^0\right) \sim N\left(\theta_{par}; \theta_{ppt}^m, \theta_{par}^{ref}\right) \tag{3}$$

In models M3 and M4, we assume participants may instead use a new central tendency (rather than their own) as prior beliefs over their partner. These are free parameters to be approximated, $\bar{\alpha}_{par}^m, \bar{\beta}_{par}^m$.

In all cases, we assume participants update their beliefs about their partner's social preferences given their partner's decisions $\boldsymbol{D}$ along trials 1–54 according to Bayes rule:

$$\theta_{par}^t = \frac{p(d^t \mid \theta_{par}; \boldsymbol{R}^t)\theta_{par}^{t-1}}{\sum_{\theta'_{par}} p(d^t \mid \theta'_{par}; \boldsymbol{R}^t)\theta'^{t-1}_{par}} \tag{4}$$

We can then marginalise over $\theta_{par}^t$ to calculate the belief participants had over their partner's social value preferences.

We assume that participants predict the partner's decision in the next trial by calculating the probability determined by the utility differences $\Delta U_{\alpha,\beta}\left(\boldsymbol{R}^{t+1}\right)$ as in phase 1, summed over the joint distribution of partner parameters, $\theta_{par}^t$:

$$\begin{aligned} p\left(\bar{d}^{t+1} = 1 \mid D^t, \boldsymbol{R}^t\right) &= \sum_{\theta_{par}} \sigma\left(\Delta U_{\alpha,\beta}\left(\boldsymbol{R}^t\right)\right) \cdot \theta_{par}^t \\ p\left(\bar{d}^{t+1} = 2 \mid D^t, \boldsymbol{R}^t\right) &= 1 - p\left(\bar{d}^{t+1} = 1 \mid D^t, \boldsymbol{R}^t\right) \end{aligned} \tag{5}$$

And then performed probability matching, so that:

$$p\left(\bar{d}^{t+1} = 1|D^t, \boldsymbol{R}^t\right) = p\left(d^{t+1} = 1|D^t, \boldsymbol{R}^t\right) \tag{6}$$

In the third phase, participants are once again asked to make choices for themselves and a new anonymous partner over 36 trials with an assumed identical utility function as in phase 1. In models M1 and M3, we assume participants use a combination of their own preferences and the posterior beliefs about their partner to form a new distribution to select between the two options available on each trial. This draws from the same formulation used previously (*Moutoussis et al., 2016*). In essence, we state that participants know their true preferences in phase 1 but are unsure about them. The inferred partner beliefs $\theta_{par}^t$ provides information to the participant about some common preference distribution both share, which in turn informs the participant's own choices $\acute{c}^t$, $t = \{1 \ldots T\}$ in the form of an adjusted belief along each dimension for phase 3, $\alpha_{ppt}$ and $\beta_{ppt}$ (*Equation 7*), using a log-likelihood of $LL = \log\left[p\left(\acute{c}^t = 1|\acute{\alpha}_{ppt}, \acute{\beta}_{ppt}, \boldsymbol{R}^t\right)\right]$. We refer to $\alpha_{ppt}$ and $\beta_{ppt}$ together as $\theta_{ppt}$ for convenience, where $\theta_{ppt}$ is a matrix over a grid of fixed values of $\alpha_{ppt}$ and $\beta_{ppt}$. To note: models M2 and M4 do not assume participants undergo this change and instead use their original phase 1 beliefs to make choices $LL = \log\left[p\left(c^t = 1 \mid \alpha_{ppt}, \beta_{ppt}, \boldsymbol{R}^t\right)\right]$.

$$\begin{aligned} p\left(\acute{\theta}_{ppt} \mid \boldsymbol{C}\right) &\sim \mathcal{N}\left(\theta_{ppt}; \acute{\theta}_{ppt}^m, \acute{\theta}_{ppt}^\sigma\right) \\ \acute{\theta}_{ppt}^\sigma &= \left(\left(\theta_{ppt}^\sigma\right)^{-2} + \left(2\left(\theta_{par}^{ref}\right)^2 + \left(\theta_{par}^\sigma\right)^2\right)^{-1}\right)^{-1} \\ \acute{\theta}_{ppt}^m &= \left(\acute{\theta}_{ppt}^\sigma\right)^2 \cdot \left[\left(\theta_{ppt}^\sigma\right)^{-2} \cdot \theta_{ppt}^m + \left(2\left(\theta_{ref}^\sigma\right)^2 + \left(\theta_{par}^\sigma\right)^2\right)^{-1} \cdot \theta_{par}^m\right] \end{aligned} \tag{7}$$

where $\theta_{par}^\sigma$ and $\theta_{par}^m$ are, respectively, the standard deviation and central tendency of the final posterior inference about the partner, $\theta_{par}^{t;54}$.

All computational models were fitted using an HBI algorithm which allows hierarchical parameter estimation while assuming random effects for group and individual model responsibility (*Piray et al., 2019*). During fitting, we added a small noise floor to distributions (2.22e$^{-16}$) before normalisation for numerical stability. Parameters were estimated using the HBI in untransformed space drawing from broad priors ($\mu_M$=0, $\sigma^2_M$=6.5, where $\boldsymbol{M}$={M1, M2, M3, M4}). This process was run independently for each group. Parameters were transformed into model-relevant space for analysis. All models and hierarchical fitting were implemented in MATLAB (version R2022B). All other analyses were conducted in R (version 4.3.3; arm64 build) running on Mac OS (Ventura 13.0). We extracted individual- and

group-level responsibility, as well as the protected exceedance probability, to assess model dominance per group.

To conduct model recovery, we simulated synthetic participants (CON = 53; BPD = 50) using their fitted parameters from the dominant model of the group (CON = M1; BPD = M4). We then performed model fitting with an identical procedure to the real behavioural data. We tested associations between model responsibility and individual parameters for the real and recovered models, as well as the association between choices and predictions made by the model from simulation and the choices and predictions made by participants in each trial.

Differences between groups for individual-level parameters were estimated using hierarchical Bayesian t-tests (*Bååth, 2014*) and hierarchical general linear models in *rStanArm*. Differences in mean between groups ($\Delta\mu$) are additionally reported with their corresponding posterior 95% high-density interval (95%HDI). Belief updates were calculated as the Kullback-Leibler Divergence between probabilities (p) from trial $t$–1 to $t$, marginalised along all possible states, $\mathbf{S}$={$s^1,s^2,\dots,s^n$}: $D_{KL}\left(P^t\|P^{t-1}\right) = \sum_s^S P^t\left(s\right)\log\frac{P^t(s)}{P^{t-1}(s)}$.

## Exploratory network analysis

To understand the individual differences of trait attributes (MZQ, RGPTSB, CTQ) with other-to-self information transfer ($\Delta\alpha_{ppt}^m$; $\Delta\beta_{ppt}^m$) across the entire sample, we performed a network analysis (*Borsboom et al., 2021*). Network analysis allows for conditional associations between variables to be estimated; each association is controlled for by all other associations in the network. It also allows for visual inspection of the conditional relationships to get an intuition for how variables are interrelated as a whole (see *Figure 2—figure supplement 6*). We implemented network analysis with the bootNet package in R using the 'estimateNetwork' function with partial correlations (*Epskamp et al., 2018*). To assess the stability of the partial correlations, we further implemented bootstrap resampling with 5000 repetitions using the 'bootnet' function. We then additionally shuffled the data and refitted the network 5000 times to determine a $p_{permuted}$ value; this indicates the probability that a conditional relationship in the original network was within the null distribution of each conditional relationship. We then performed FDR correction on the resulting p-values. We additionally controlled for group status for all variables in a supplementary analysis (*Supplementary file 4*).

## Acknowledgements

We would like to greatly thank all participants who took part in the research. JMB is supported by a Wellcome Trust award (228268/Z/23/Z) and as a scholar within the FENS-Kavli Network of Excellence. Funding for PD was from the Max Planck Society and the Humboldt Foundation. PD is a member of the Machine Learning Cluster of Excellence, EXC number 2064/1 – Project number 39072764 and of the Else Kroner Medical Scientist College 'ClinbrAIn: Artificial Intelligence for Clinical Brain Research'.

## Additional information

### Funding

| Funder | Grant reference number | Author |
| --- | --- | --- |
| Wellcome Trust | 228268/Z/23/Z | Joseph M Barnby |
| FENS-Kavli Network of Excellence | | Joseph M Barnby |
| Alexander von Humboldt-Stiftung | | Peter Dayan |
| Else Kroner Medical Scientist College | | Peter Dayan |
| Max Planck Society | | Peter Dayan |
| Machine Learning Cluster of Excellence | 2064/1 | Peter Dayan |

| Funder | Grant reference number | Author |
| --- | --- | --- |

The funders had no role in study design, data collection and interpretation, or the decision to submit the work for publication. For the purpose of Open Access, the authors have applied a CC BY public copyright license to any Author Accepted Manuscript version arising from this submission.

## Author contributions

Joseph M Barnby, Conceptualization, Data curation, Software, Formal analysis, Supervision, Visualization, Methodology, Writing – original draft, Project administration, Writing – review and editing; Jen Nguyen, Investigation, Methodology, Writing – original draft; Julia Griem, Conceptualization, Resources, Investigation, Project administration, Writing – review and editing; Magdalena Wloszek, Project administration; Henry Burgess, Software, Writing – review and editing; Linda J Richards, P Read Montague, Resources, Writing – review and editing; Jessica Kingston, Supervision, Writing – review and editing; Gavin Cooper, Validation, Writing – review and editing; Peter Dayan, Conceptualization, Formal analysis, Writing – review and editing; Tobias Nolte, Conceptualization, Resources, Supervision, Project administration, Writing – review and editing; Peter Fonagy, Conceptualization, Resources, Supervision, Writing – review and editing; London Personality and Mood Disorders Consortium, Consortium support

## Author ORCIDs

Joseph M Barnby ⓘ https://orcid.org/0000-0001-6002-1362
Julia Griem ⓘ http://orcid.org/0000-0002-1779-5255
Henry Burgess ⓘ https://orcid.org/0000-0002-3481-952X
Linda J Richards ⓘ https://orcid.org/0000-0002-7590-7390
Peter Dayan ⓘ https://orcid.org/0000-0003-3476-1839
Tobias Nolte ⓘ http://orcid.org/0000-0002-6834-7727
Peter Fonagy ⓘ http://orcid.org/0000-0003-0229-0091

## Ethics

Human subjects: Participants were recruited for a larger study investigating social exchanges in BPD and Anti-Social Personality Disorder (approved by the Research Ethics Committee for Wales, 12/WA/0283; informed consent and consent to publish was obtained).

Reviewer #1 (Public review): https://doi.org/10.7554/eLife.104008.4.sa1
Author response https://doi.org/10.7554/eLife.104008.4.sa2

# Additional files

## Supplementary files

Supplementary file 1. Supplementary Table 1, SVO Choices. Option pair rewards for each phase and their corresponding 'type'. Within phase, the order of trials was randomised. P=Prosocial, I=Individualistic, C=Competitive. S1=reward to self for option 1. S2=reward to self for option 2. O1=reward to other for option 1. O2=reward to other for option 2.

Supplementary file 2. Supplementary Table 2, Individual Model Parameters. Model parameters of M1-M4 following independent hierarchical fitting for all participants.

Supplementary file 3. Supplementary Table 3, Individual Parameter Relationships. Random-effect linear relationships between $D_{KL}$, trial, group, and preferences type for each model (M1-M4) following independent hierarchical fitting for all participants. Estimates are the scaled change in $D_{KL}$ as a result of each fixed effect. ID was used as a random variable to control for within-subject effects. Group effects (CON vs borderline personality disorder [BPD]) were analysed for the $D_{KL}$ within each preference type.

Supplementary file 4. Supplementary Table 4, Network Correlations. Bootstrapped results with their 95% CI with and without group status regressed against psychometric variables.

Supplementary file 5. Reaction times in the Intentions Game.

MDAR checklist

## Data availability

All data and corresponding code can be found here: https://github.com/josephmbarnby/Social-Transfer_Barnby_etal_2024 (copy archived at *Barnby, 2025*).

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
