## [Editor Report · eLife Assessment]

The findings are **important** and intriguing, with theoretical or practical implications beyond a single subfield. The computational methods employed are clever and sophisticated and the strength of evidence is **convincing**. Both the hypotheses and the exploratory nature of additional analyses are clearly stated.

---

## [Referee Report · Reviewer #1 (Public review)]

Summary:

The authors use a sophisticated and novel task design and Bayesian computational modeling to test their hypothesis that information generalization (operationalized as a combination of self-insertion and social contagion) in social situations is disrupted in Borderline Personality Disorder. Their main finding relates to the observation that two different models best fit the two tested groups: While the model assuming both self-insertion and social contagion to be present when estimating others' social value preferences fit the control group best, a model assuming neither of these processes provided the best fit to BPD participants.

Strengths:

The two revisions have substantially strengthened the paper and the manuscript is much clearer and easier to follow now. The introduction now precisely states the author's hypotheses, and the connections to the theoretical framework are presented with much greater clarity. I appreciate that the authors now clearly label exploratory analyses where applicable.

The strengths of the presented work lie in the sophisticated task design and the thorough investigation of their theory by use of mechanistic computational models to elucidate social decision-making and learning processes in BPD. Although at present it is not clear whether the differing strategies in impression formation observed in BPD are in any way causal to negative outcomes in the condition, the study represents an important step towards better understanding cognitive processes in BPD. The paradigm and models are also potentially relevant for the investigation of other psychiatric conditions.

---

## [Author Response]

The following is the authors’ response to the previous reviews.

**Reviewer 1:**
The authors frequently refer to their predictions and theory as being causal, both in the manuscript and in their response to reviewers. However, causal inference requires careful experimental design, not just statistical prediction. For example, the claim that "algorithmic differences between those with BPD and matched healthy controls" are "causal" in my opinion is not warranted by the data, as the study does not employ experimental manipulations or interventions which might predictably affect parameter values. Even if model parameters can be seen as valid proxies to latent mechanisms, this does not automatically mean that such mechanisms cause the clinical distinction between BPD and CON, they could plausibly also refer to the effects of therapy or medication. I recommend that such causal language, also implicit to expressions like "parameter influences on explicit intentional attributions", is toned down throughout the manuscript.

Thankyou for this chance to be clearer in the language. Our models and paradigm introduce a from of temporal causality, given that latent parameter distributions are directly influenced by latent parameter estimates at a previous point in time (self-uncertainty and other uncertainty directly governs social contagion). Nevertheless, we appreciate the reviewers perspective and have now toned down the language to reflect this.

Abstract:

‘Our model makes clear predictions about the mechanisms of social information generalisation concerning both joint and individual reward.’

Discussion:

‘We can simulate this by modelling a framework that incorporates priors based on both self and a strong memory impression of a notional other (Figure S3).’

‘We note a strength of this work is the use of model comparison to understand algorithmic differences between those with BPD and matched healthy controls.’

Although the authors have now much clearer outlined the stuy's aims, there still is a lack of clarity with respect to the authors' specific hypotheses. I understand that their primary predictions about disruptions to self-other generalisation processes underlying BPD are embedded in the four main models that are tested, but it is still unclear what specific hypotheses the authors had about group differences with respect to the tested models. I recommend the authors specify this in the introduction rather than refering to prior work where the same hypotheses may have been mentioned.

Thankyou for this further critique which has enabled us to more cleary refine our introduction. We have now edited our introduction to be more direct about our hypotheses, that these hypotheses are instantiated into formal models, and what our predictions were. We have also included a small section on how previous predictions from other computational assessments of BPD link to our exploratory work, and highlighted this throughout the manuscript.

‘This paper seeks to address this gap by testing explicitly how disruptions in self-other generalization processes may underpin interpersonal disruptions observed in BPD. Specifically, our hypotheses were: (i) healthy controls will demonstrate evidence for both self-insertion and social contagion, integrating self and other information during interpersonal learning; and (ii) individuals with BPD will exhibit diminished self-other integration, reflected in stronger evidence for observations that assume distinct self-other representations.

We tested these hypotheses by designing a dynamic, sequential, three-phase Social Value Orientation (Murphy & Ackerman, 2014) paradigm—the Intentions Game—that would provide behavioural signatures assessing whether BPD differed from healthy controls in these generalization processes (Figure 1A). We coupled this paradigm with a lattice of models (M1-M4) that distinguish between self-insertion and social contagion (Figure 1B), and performed model comparison:

M1. Both self-to-other (self-insertion) and other-to-self (social contagion) occur before and after learningM2. Self-to-other transfer only occursM3. Other-to-self transfer only occursM4. Neither transfer process, suggesting distinct self-other representations

We additionally ran exploratory analysis of parameter differences and model predictions between groups following from prior work demonstrating changes in prosociality (Hula et al., 2018), social concern (Henco et al., 2020), belief stability (Story et al., 2024a), and belief updating (Story, 2024b) in BPD to understand whether discrepancies in self-other generalisation influences observational learning. By clearly articulating our hypotheses, we aim to clarify the theoretical contribution of our findings to existing literature on social learning, BPD, and computational psychiatry.’

Caveats should also be added about the exploratory nature of the many parameter group comparisons. If there are any predictions about group differences that can be made based on prior literature, the authors should make such links clear.

Thank you for this. We have now included caveats in the text to highlight the exploratory nature of these group comparisons, and added direct links to relevant literature where able:

Introduction

‘We additionally ran exploratory analysis of parameter differences and model predictions between groups following from prior work demonstrating changes in prosociality (Hula et al., 2018), social concern (Henco et al., 2020), belief stability (Story et al., 2024a), and belief updating (Story, 2024b) in BPD to understand whether discrepancies in self-other generalisation influences observational learning. By clearly articulating our hypotheses, we aim to clarify the theoretical contribution of our findings to existing literature on social learning, BPD, and computational psychiatry.’

Model Comparison

‘We found that CON participants were best fit at the group level by M1 (Frequency = 0.59, Exceedance Probability = 0.98), whereas BPD participants were best fit by M4 (Frequency = 0.54, Exceedance Probability = 0.86; Figure 2A). This suggests CON participants are best fit by a model that fully integrates self and other when learning, whereas those with BPD are best explained as holding disintegrated and separate representations of self and other that do not transfer information back and forth.

We first explore parameters between separate fits (see Methods). Later, in order to assuage concerns about drawing inferences from different models, we examined the relationships between the relevant parameters when we forced all participants to be fit to each of the models (in a hierarchical manner, separated by group). In sum, our model comparison is supported by convergence in parameter values when comparisons are meaningful (see Supplementary Materials). We refer to both types of analysis below.’

Phase 2 analysis

‘Prior work predicts those with BPD should focus more intently on public social information, rather than private information that only concerns one party (Henco et al., 2020). In BPD participants, only new beliefs about the relative reward preferences – mutual outcomes for both player - of partners differed (see Fig 2E): new median priors were larger than median preferences in phase 1 (mean \begin{document}$\left[\beta_{\text {par }}^{m}\right]$\end{document} = -0.47; \begin{document}$\Delta \mu\left[\beta_{p p t}^{m}-\beta_{p a r}^{m}\right]$\end{document} = -6.10, 95%HDI: -7.60, -4.60).’

‘Models of moral preference learning (Story et al., 2024) predicts that BPD vs non-BPD participants have more rigid beliefs about their partners. We found that BPD participants were equally flexible around their prior beliefs about a partner’s relative reward preferences (\begin{document}$\Delta \mu\left[\beta_{\text {par }}^{\text {ref }}\right]$\end{document} = -1.60, 95%HDI: -3.42, 0.23), and were less flexible around their beliefs about a partner’s absolute reward preferences (\begin{document}$\Delta \mu\left[\alpha_{\text {par }}^{\text {ref }}\right]$\end{document} = -4.09, 95%HDI: -5.37, -2.80), versus CON (Figure 2B).’

Phase 3 analysis

‘Prior work predicts that human economic preferences are shaped by observation (Panizza, et al., 2021; Suzuki et al. 2016; Yu et al, 2021), although little-to-no work has examined whether contagion differs for relative vs. absolute preferences. Associative models predict that social contagion may be exaggerated in BPD (Ereira et al., 2018).… As a whole, humans are more susceptible to changing relative preferences more than selfish, absolute reward preferences, and this is disrupted in BPD.’

Psychometric and Intentional Attribution analysis

‘Childhood trauma, persecution, and poor mentalising in BPD are all predicted to disrupt one’s ability to change (Fonagy & Luyten, 2009).’

‘Prior work has also predicted that partner-participant preference disparity influences mental state attributions (Barnby et al., 2022; Panizza et al., 2021).’

I'm not sure I understand why the authors, after adding multiple comparison correction, now list two kinds of p-values. To me, this is misleading and precludes the point of multiple comparison corrections, I therefore recommend they report the FDR-adjusted p-values only. Likewise, if a corrected p-value is greater than 0.05 this should not be interpreted as a result.

We have now adjusted the exploratory results to include only the FDR corrected values in the text.

‘We assessed conditional psychometric associations with social contagion under the assumption of M3 for all participants. We conducted partial correlation analyses to estimate relationships conditional on all other associations and retained all that survived bootstrapping (5000 reps), permutation testing (5000 reps), and subsequent FDR correction. When not controlled for group status, RGPTSB and CTQ scores were both moderately associated with MZQ scores (RGPTSB r = 0.41, 95%CI: 0.23, 0.60, p[fdr]=0.043; CTQ r = 0.354 95%CI: 0.13, 0.56, p[fdr]=0.02). This was not affected by group correction. CTQ scores were moderately and negatively associated with shifts in individualistic reward preferences (\begin{document}$\Delta \alpha_{p p t}^{m}$\end{document}; r = -0.25, 95%CI: -0.46, -0.04, p[fdr]=0.03). This was not affected by group correction. MZQ scores were in turn moderately and negatively associated with shifts in prosocial-competitive preferences (\begin{document}$\Delta \beta_{p p t}^{m}$\end{document}) between phase 1 and 3 (r = -0.26, 95%CI: -0.46, -0.06, p[fdr]=0.03). This was diminished when controlled for group status (r = 0.13, 95%CI: -0.34, 0.08, p[fdr]=0.20). Together this provides some evidence that self-reported trauma and self-reported mentalising influence social contagion (Fig S11). Social contagion under M3 was highly correlated with contagion under M1 demonstrating parsimony of outcomes across models (Fig S12).

Prior work has predicted that partner-participant preference disparity influences mental state attributions (Barnby et al., 2022; Panizza et al., 2021). We tested parameter influences on explicit intentional attributions in Phase 2 while controlling for group status. Attributions included the degree to which they believed their partner was motived by harmful intent (HI) and self-interest (SI). According with prior work (Barnby et al., 2022), greater disparity of absolute preferences before learning was associated on a trend level with reduced attributions of SI (<\begin{document}$\rho\left[\left|\alpha_{p a r}^{m}-\alpha_{p p t}^{m}\right|\right]$\end{document} = -0.23, p[fdr]=0.08), and greater disparity of relative preferences before learning exaggerated attributions of HI \begin{document}$\rho\left[\left|\beta_{p a r}^{m}-\beta_{p p t}^{m}\right|\right]$\end{document} = 0.21, p[fdr] =0.08, but did not survive correction (Figure S4B). This is likely due to partners being significantly less individualistic and prosocial on average compared to participants (\begin{document}$\Delta \mu[\alpha]$\end{document} = -5.50, 95%HDI: -7.60, -3.60; \begin{document}$\Delta \mu[\beta]$\end{document} = 12, 95%HDI: 9.70, 14.00); partners are recognised as less selfish and more competitive.’

Can the authors please elaborate why the algorithm proposed to be employed by BPD is more 'entropic', especially given both their self-priors and posteriors about partners' preferences tended to be more precise than the ones used by CON? As far as I understand, there's nothing in the data to suggest BPD predictions should be more uncertain. In fact, this leads me to wonder, similarly to what another reviewer has already suggested, whether BPD participants generate self-referential priors over others in the same way CON participants do, they are just less favourable (i.e., in relation to oneself, but always less prosocial) - I think there is currently no model that would incorporate this possibility? It should at least be possible to explore this by checking if there is any statistical relationship between the estimated θ_ppt^m and 〖p(θ〗_par |D^0).

Thank you for this opportunity to be clearer in our wording. We belief the reviewer is referring to this line in the discussion: ‘In either case, the algorithm underlying the computational goal for BPD participants is far higher in entropy and emphasises a less stable or reliable process of inference.’

We note in the revised Figure 2 panel E and in the results that those with BPD under M4 show insertion along absolute reward (they still expect diminished selfishness in others), but neutral priors over relative reward (around 0, suggesting expectations of neither prosocial or competitive tendencies of others). Thus, θ_ppt^m (self preference) and θ_par^m (other preference) are tightly associated for absolute, but not relative reward.

In our wording, we meant that whether under model M4 or M1, those with BPD either show a neutral prior over relative reward (M4) or a prior with large variance over relative reward (M1), showing expectations of difference between themselves and their partner. In both cases, expectation about a partner’s absolute reward preferences is diminished vs. CON participants. We have strengthened our language in the discussion to clarify this:

‘In either case, the algorithm underlying the computational goal for BPD participants is far higher in uncertainty, whether through a neutral central tendency (M4) or large variance (M1) prior over relative reward in phase 2, and emphasises a less certain and reliable expectation about others.’

To note, social contagion under M3 was highly correlated with contagion under M1 (see Fig S11). This provides some preliminary evidence that trauma impacts beliefs about individualism directly, whereas trauma and persecutory beliefs impact beliefs about prosociality through impaired trait mentalising" - I don't understand what the authors mean by this, can they please elaborate and add some explanation to the main text?

We have now clarified this in the text:

‘Together this provides some evidence that self-reported trauma and self-reported mentalising influence social contagion (Fig S11). Social contagion under M3 was highly correlated with contagion under M1 demonstrating parsimony of outcomes across models (Fig S12).’

I noted that at least some of the newly added references have not been added to the bibliography (e.g., Hitchcock et al. 2022).

Thankyou for noticing this omission. We have now ensured all cited works are in the reference list.

**Reviewer 2:**
The paper is not based on specific empirical hypotheses formulated at the outset, but, rather, it uses an exploratory approach. Indeed, the task is not chosen in order to tackle specific empirical hypotheses. This, in my view, is a limitation since the introduction reads a bit vague and it is not always clear which gaps in the literature the paper aims to fill. As a further consequence, it is not always clear how the findings speak to previous theories on the topic.’As I wrote in the public review, however, I believe that an important limitation of this work is that it was not based on testing specific empirical hypotheses formulated at the outset, and on selecting the experimental paradigm accordingly. This is a limitation because it is not always clear which gaps in the literature the paper aims to fill. As a consequence, although it has improved substantially compared to the previous version, the introduction remains a bit vague. As a further consequence, it is not always clear how the findings speak to previous theories on the topic. Still, despite this limitation, the paper has many strengths, and I believe it is now ready for publication

Thank you for this further critique. We appreciate your appraisal that the work has improved substantially and is ready for publication. We nevertheless have opted to clarify our introduction and aprior predictions throughout the manuscript (please see response to Reviewer 1).

**Reviewer 3:**
Although the authors note that their approach makes "clear and transparent a priori predictions," the paper could be improved by providing a clear and consolidated statement of these predictions so that the results could be interpreted vis-a-vis any a priori hypotheses.

In line with comments from both Reviewer 1 and 2, we have clarified our introduction to make it clear what our aprior predictions and hypotheses are about our core aims and exploratory analyses (see response to Reviewer 1).

The approach of using a partial correlation network with bootstrapping (and permutation) was interesting, but the logic of the analysis was not clearly stated. In particular, there are large group (Table 1: CON vs. BPD) differences in the measures introduced into this network. As a result, it is hard to understand whether any partial correlations are driven primarily by mean differences in severity (correlations tend to be inflated in extreme groups designs due to the absence of observation in middle of scales forming each bivariate distribution). I would have found these exploratory analyses more revealing if group membership was controlled for.

Thank you for this chance to be clearer in our methods. We have now written a more direct exposition of this exploratory method:

‘Exploratory Network Analysis

To understand the individual differences of trait attributes (MZQ, RGPTSB, CTQ) with other-to-self information transfer (\begin{document}$\Delta \alpha_{p p t}^{m}; \Delta \beta_{p p t}^{m}$\end{document}) across the entire sample we performed a network analysis (Borsboom, 2021). Network analysis allows for conditional associations between variables to be estimated; each association is controlled for by all other associations in the network. It also allows for visual inspection of the conditional relationships to get an intuition for how variables are interrelated as a whole (see Fig S11). We implemented network analysis with the bootNet package in r using the ‘estimateNetwork’ function with partial correlations (Epskamp, Borsboom & Fried, 2018). To assess the stability of the partial correlations we further implemented bootstrap resampling with 5000 repetitions using the ‘bootnet’ function. We then additionally shuffled the data and refitted the network 5000 times to determine a *ppermuted* value; this indicates the probability that a conditional relationship in the original network was within the null distribution of each conditional relationship. We then performed False Discovery Rate correction on the resulting p-values. We additionally controlled for group status for all variables in a supplementary analysis (Table S4).’

We have also further corrected for group status and reported these results as a supplementary table, and also within the main text alongside the main results. We have opted to relegate Figure 4 into a supplementary figure to make the text clearer.

‘We explored conditional psychometric associations with social contagion under the assumption of M3 for all participants (where everyone is able to be influenced by their partner). We conducted partial correlation analyses to estimate relationships conditional on all other associations and retained all that survived bootstrapping (5000 reps), permutation testing (5000 reps), and subsequent FDR correction. When not controlled for group status, RGPTSB and CTQ scores were both moderately associated with MZQ scores (RGPTSB r = 0.41, 95%CI: 0.23, 0.60, p[fdr]=0.043; CTQ r = 0.354 95%CI: 0.13, 0.56, p[fdr]=0.02). This was not affected by group correction. CTQ scores were moderately and negatively associated with shifts in individualistic reward preferences (\begin{document}$\Delta \alpha_{p p t}^{m}$\end{document}; r = -0.25, 95%CI: -0.46, -0.04, p[fdr]=0.03). This was not affected by group correction. MZQ scores were in turn moderately and negatively associated with shifts in prosocial-competitive preferences (\begin{document}$\Delta \beta_{p p t}^{m}$\end{document}) between phase 1 and 3 (r = -0.26, 95%CI: -0.46, -0.06, p[fdr]=0.03). This was diminished when controlled for group status (r = 0.13, 95%CI: -0.34, 0.08, p[fdr]=0.20). Together this provides some evidence that self-reported trauma and self-reported mentalising influence social contagion (Fig S11). Social contagion under M3 was highly correlated with contagion under M1 demonstrating parsimony of outcomes across models (Fig S12).’

Discussion first para: "effected -> affected"

Thanks for spotting this. We have now changed it.

Add "s" to "participant: "Notably, despite differing strategies, those with BPD achieved similar accuracy to CON participant."

We have now changed this.